# Topography-based statistical modelling reveals high spatial variability and seasonal emission patches in forest floor methane flux

Elisa Vainio[1,2,3], Olli Peltola[4], Ville Kasurinen[1], Antti-Jussi Kieloaho[5], Eeva-Stiina Tuittila[6], Mari Pihlatie[1,2,7]

[1] Institute for Atmospheric and Earth System Research (INAR) / Physics, University of Helsinki, Finland
[2] Environmental Soil Sciences, Department of Agricultural Sciences, University of Helsinki, Finland
[3] Institute for Atmospheric and Earth System Research (INAR) / Forest Science, University of Helsinki, Finland
[4] Climate Research Programme, Finnish Meteorological Institute, Helsinki, Finland
[5] Natural Resources Institute Finland (LUKE), Helsinki, Finland
[6] School of Forest Sciences, University of Eastern Finland, Joensuu, Finland
[7] Viikki Plant Science Centre (ViPS), University of Helsinki, Finland

*Correspondence to*: Elisa Vainio (elisa.vainio@helsinki.fi)

**Abstract.** Boreal forest soils are globally an important sink for methane ($CH_4$), while these soils are also capable to emit $CH_4$ under favourable conditions. Soil wetness is a well-known driver of $CH_4$ flux, and the wetness can be estimated with several terrain indices developed for the purpose. The aim of this study was to quantify the spatial variability of the forest floor $CH_4$ flux with a topography-based upscaling method connecting the flux with its driving factors. We conducted spatially extensive forest floor $CH_4$ flux and soil moisture measurements, complemented with ground vegetation classification, in a boreal pine forest. We then modelled the soil moisture with a Random Forest model using digital-elevation-model-derived topographic indices, based on which we upscaled the forest floor $CH_4$ flux. The modelling was performed for two seasons: May–July and August–October. Additionally, we evaluated the number of flux measurement points needed to get an accurate estimate of the flux at the whole study site merely by averaging. Our results demonstrate high spatial heterogeneity in the forest floor $CH_4$ flux resulting from the soil moisture variability, as well as from the related ground vegetation. The mean measured $CH_4$ flux at the sample points was −5.07 µmol m$^{-2}$ h$^{-1}$ in May–July and −8.67 µmol m$^{-2}$ h$^{-1}$ in August–October, while the modelled flux for the whole area was −7.42 and −9.91 µmol m$^{-2}$ h$^{-1}$ for the two seasons, respectively. The spatial variability in the soil moisture and consequently in the $CH_4$ flux was higher in the early summer (modelled range from −12.3 to 6.19 µmol m$^{-2}$ h$^{-1}$) compared to the autumn period (range from −14.6 to −2.12 µmol m$^{-2}$ h$^{-1}$), and overall the $CH_4$ uptake rate was higher in autumn compared to early summer. In the early summer there were patches emitting high amounts of $CH_4$, however, these wet patches got drier and smaller in size towards the autumn, changing their dynamics to $CH_4$ uptake. The mean values of the measured and modelled $CH_4$ flux for the sample point locations were similar, indicating that the model was able to reproduce the results. For the whole site, upscaling predicted stronger $CH_4$ uptake compared to simply averaging over the sample points. The results highlight the small-scale spatial variability of the boreal forest floor $CH_4$ flux, and the importance of soil chamber placement in order to obtain spatially representative $CH_4$ flux results. To predict the $CH_4$ fluxes over large areas more reliably, the locations of the sample points should be selected based on the spatial variability of the driving parameters, in addition to linking the measured fluxes with the parameters.

## 1 Introduction

Methane ($CH_4$) is an important and strong greenhouse gas, of which largest natural source to the atmosphere is
wetlands (Kirschke et al., 2013; Saunois et al., 2016). While oxidation by hydroxyl radicals (OH) in the atmosphere
form the largest natural $CH_4$ sink, boreal upland forests are considered as a globally important terrestrial sink due to
soil $CH_4$ oxidation by methanotrophs (Kirschke et al., 2013; Saunois et al., 2016). The sink role of upland forests is
well in agreement with the current paradigm where methanotrophy only occurs in oxic conditions, while
methanogenesis requires anoxic conditions. However, $CH_4$ producing methanogens are found to be universal also in
well-drained upland soils (Angel et al., 2012), which is linked to the findings that methanogenesis can occur in
anaerobic microenvironments within oxic soils (Angel et al., 2011; Angle et al., 2017).

As the availability of oxygen is the main controller for $CH_4$ dynamics, soil moisture and water table level are among
the most important factors regulating $CH_4$ formation, as well as consumption, in soils. When soils become inundated
with water, the environment often turns anoxic, thus creating favourable conditions for methanogenesis – however,
there are likely notable time lags between the start of inundation and methanogenesis, complicating the analyses of
dependencies between these processes. Consequently, upland boreal forest soils (Lohila et al., 2016; Matson et al.,
2009; Savage et al., 1997), and even the whole forest ecosystems (Shoemaker et al., 2014), can shift from $CH_4$
consumption to $CH_4$ emission, or vice versa, following the soil water conditions. Besides soil moisture, temperature
is known to be an important factor in regulating $CH_4$ fluxes, by controlling several microbial reactions, including
methanogenesis and methanotrophy (Luo et al., 2013; Praeg et al., 2017; Yvon-Durocher et al., 2014). Similarly to
microbial production of $CH_4$, non-microbial $CH_4$ production in soil (Jugold et al., 2012; Wang et al., 2013a) has also
been linked to soil water conditions and temperature: the alternation of soil drying and re-wetting (Jugold et al., 2012),
as well as high temperature (Jugold et al., 2012; B. Wang et al., 2013) enhances the non-microbial $CH_4$ emissions.

Spahni et al. (2011) estimated global $CH_4$ emissions from occasionally wet mineral soils to be 58–93 Tg $CH_4$ year$^{-1}$,
accounting for 11–18% of the global emissions (depending on the scenario). Annual $CH_4$ flux of upland sites is
evaluated to range from –23 to 73 g $CH_4$ m$^{-2}$ year$^{-1}$ (Treat et al., 2018). Nevertheless, aerated soils are generally
considered to consume $CH_4$, while $CH_4$ production via methanogenesis in occasionally wet mineral soils is neglected
from most of the global models (Curry, 2007; Saunois et al., 2016). Furthermore, the division of ecosystems into
upland and wetland sites is to some extent imprecise, and thus subject to continuous discussion, as the intrinsic
definition of 'upland' may vary from study to study. Usually the concept of 'upland' is relative to the surrounding
topography, and there is no uniform limit for e.g. minimum elevation.

The upland forest $CH_4$ emission estimates are partly based on observations above forest canopies (Flanagan et al.,
2020; Mikkelsen et al., 2011; Shoemaker et al., 2014). They further raise the question whether these $CH_4$ emissions
originate only from the forest floor, or do trees, which have also been reported to emit $CH_4$ (e.g. Gauci et al. 2010;
Machacova et al. 2016) contribute to the ecosystem level flux. As the sources and sinks within the ecosystems are
not adequately known or accounted, forest floor $CH_4$ fluxes require revisit and thorough estimation in all the climatic
zones, especially in the boreal zone where the climate warming is pronounced, and both at 'upland' and 'lowland'
sites with an emphasized focus on the local topography.

In order to precisely estimate the forest floor $CH_4$ flux variation, determining the variability of the driving parameters,
i.e. particularly soil moisture, is needed. Airborne lidar (*light detection and ranging*) is an active remote sensing
method that can be used to observe the vegetation and terrain (Jaboyedoff et al., 2012), and which is very effective
in forests (Korpela et al., 2009). Soil moisture is highly dependent on the terrain topography, like elevation and slope,

and there are several digital elevation model (DEM)-derived digital terrain indices developed for estimating soil wetness (Ågren et al., 2014). When combining lidar-based measurements to the variables of interest measured onsite, it is possible to create landscape-scale maps of the studied variable, such as forest floor/soil $CH_4$ exchange (Kaiser et al., 2018; Sundqvist et al., 2015; Warner et al., 2019) or soil moisture (Kemppinen et al., 2018).

In this study, we used relatively high number of measurement points (60 points on an area of ca. 10 ha) in order to fully cover the small-scale spatial variability in the $CH_4$ flux and its driving forces. Similar type of studies using chamber measurements are rarely based on more than 20 measurement points, yet assuming that they are representative for a larger area. The aim of this study was 1) to quantify the spatial variation in the forest floor $CH_4$ exchange, 2) to quantitatively link small-scale spatial variability in the upland forest floor $CH_4$ exchange to the topography, soil moisture and vegetation structure, and 3) to detect the potential $CH_4$-emitting patches (hot spots). We combined the $CH_4$ flux data with the driving parameters to produce an upscaled ecosystem-scale forest floor $CH_4$ flux of the area. Only a few studies have applied similar approach (Kaiser et al., 2018; Warner et al., 2019), of which Kaiser et al. (2018) in boreal coniferous forest, emphasizing the novelty of this research.

## 2 Materials and methods

### 2.1 Site description and experimental design

In order to quantify the spatial variability, we conducted forest floor $CH_4$ flux and soil moisture measurements at 60 sample points covering an area of ca. 10 hectares around the SMEAR II station (*Station for Measuring Ecosystem-Atmosphere Relations*) in Hyytiälä, southern Finland (61° 51' N, 24° 17' E; 160–180 m a.s.l.). The station is a combined ecosystem and atmospheric site in the ICOS (*Integrated Carbon Observation System*) network. The measurements were performed during two growing seasons (2013 and 2014). The site has been regenerated in 1962 by prescribed burning and sowing *Pinus sylvestris* (Scots pine) (Hari and Kulmala, 2005). The mineral soils at the area are mostly podzols, while there are also some small peaty depressions, and some areas with almost no topsoil on the bedrock (Ilvesniemi et al., 2009). The soil at the site is rather shallow (5–150 cm) on top of the bedrock (Hari and Kulmala, 2005). Annual mean temperature and precipitation in 1981–2010 have been 3.5 °C and 711 mm, respectively (Pirinen et al., 2012).

In addition to *P. sylvestris* as a dominating tree species, prevalent species at the site are *Picea abies* (Norway spruce), *Betula pendula* (silver birch), *Betula pubescens* (downy birch), together with some *Juniperus communis*, *Salix* sp., and *Sorbus aucuparia* (Ilvesniemi et al., 2009). The ground vegetation is mainly composed of *Vaccinium myrtillus* (European blueberry) and *Vaccinium vitis-idaea* (lingonberry), together with e.g. *Deschampsia flexuosa*, *Trientalis europaea*, *Maianthemum bifolium*, *Linnaea borealis*, and *Calluna vulgaris* (Ilvesniemi et al., 2009). The most common mosses are *Pleurozium schreberi*, *Dicranum polysetum*, *Polytrichum* sp., *Hylocomium splendens*, and *Sphagnum* sp. (Ilvesniemi et al., 2009).

To represent the heterogeneity in vegetation and soil moisture we located six sample points on the highest area on top of the hill, and 54 at all the wind directions from the hilltop (Fig. 1). The sample points were identified based on the cardinal and intermediate directions from the centre of the studied area (the main mast of SMEAR II), thus having eight sectors (north–north-east sector N–NE, north-east–east NE–E, etc.), accompanied by an Arabic numeral (1–9) depending on the distance from the centre of the study area (e.g. sample point SE–S-1 being the closest to the centre

at the sector SE–S). The hilltop sample points are located at the sectors N–NE, NE–E, and E–SE, but they are labelled with the letter H instead of the directions.

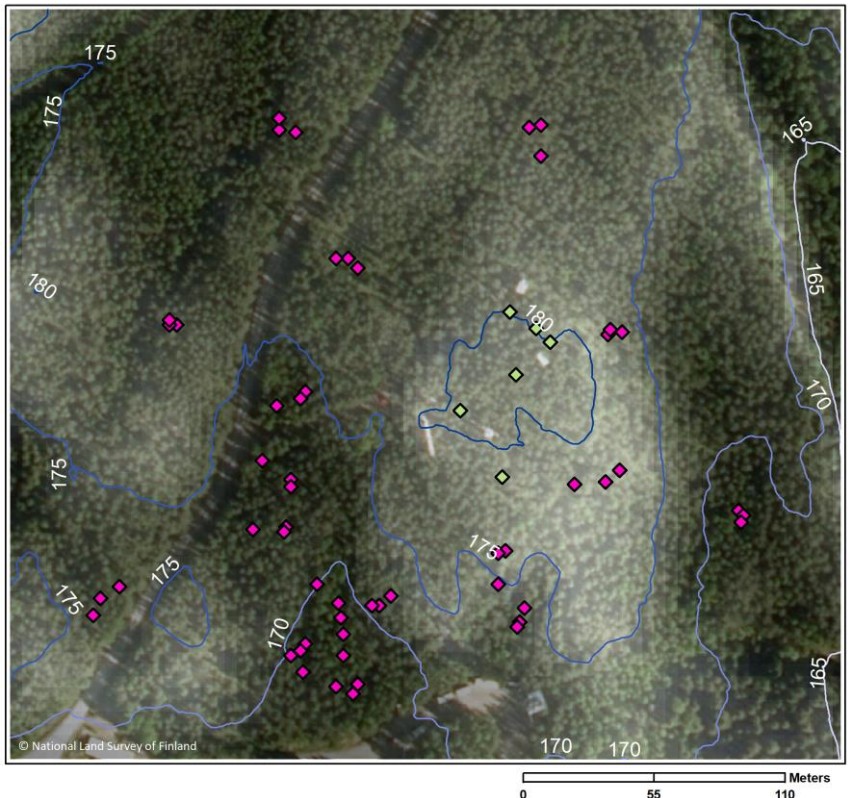

**Figure 1. Locations of the sample points (diamonds) at the study site. The hilltop sample points are coloured light green, and the rest are pink. The cartographic depth-to-water index (DTW) is showed on top of the aerial image, lighter colour**
**indicating higher DTW, i.e. drier soil. (Copyright of the map: National Land Survey of Finland.)**

### 2.2 Flux measurements

The flux measurements were conducted with *non-steady-state non-flow-through* static chambers (Livingston and Hutchinson, 1995) at each sample point, principally following the guidelines compiled in the ICOS protocol (Pavelka et al., 2018). The majority of the measurements were conducted with opaque aluminium or stainless steel chambers.
The hilltop chambers were on average 0.027 $m^3$ including the collar (depending on the collar height and the vegetation inside the chamber), covering a forest floor area of $0.40 \times 0.29$ m. The rest of the chambers were on average 0.102 $m^3$ covering an area of 0.55 m x 0.55 m. Part of the measurements were conducted with transparent chambers made of FEP (fluorinated ethylene propylene) foil and PTFE (polytetrafluoroethylene) tape, in order to test the effect of photosynthetically active radiation (PAR) on the $CH_4$ flux. The transparent chamber measurements were always
performed 15–155 minutes before the opaque chamber at the same sample points. As there was no significant difference in the flux between the chamber types, the data were merged (transparent chambers were used in 14% of the measurements in the final data). All the chambers were equipped with a fan to ensure mixing of the chamber headspace air, and a vent-tube to minimize pressure disturbances inside the chamber. The collars were installed in May 2013 one week before the beginning of the measurements (except for the hilltop chambers, which were installed
already in 2002 (Pihlatie et al., 2007)) at the depth of ca. 5 cm to avoid cutting of tree roots and to minimize the sideways diffusion in the soil affecting the flux (Hutchinson and Livingston, 2001). Fine quartz sand was added to the edges of the collars to ensure the installation.

The chambers were closed for 35–45 minutes and 5 samples were taken during each closure. Small part of the closures (10% of the final data) were 75 min with 7 samples, due to separate study on $N_2O$ fluxes. The samples were taken with 65 ml syringes (BD Plastipak™, Becton, Dickinson and Company, New Jersey, USA), and samples of 20 ml were inserted into glass vials (12 ml, Labco Exetainer®, Labco Limited, Wales, UK) after flushing the vial with the sample. The samples were stored in dark at +5 °C before analyses with a gas chromatograph (GC) (7890A, Agilent Technologies, California, USA) equipped with a flame ionization detector (for details see Pihlatie et al. 2013). The chamber headspace air temperature was also recorded (DT-612, CEM Instruments, Shenzhen Everbest Machinery Industry Co. Ltd., Shenzhen, China) during the measurements for the flux calculation.

Measurements from the hilltop sample points were conducted every 2–9 weeks between 21 March and 20 December 2013, and every 2–7 weeks between 10 April and 27 November 2014. The other 54 sample points were measured on average every 3–4 weeks between 29 May–13 September 2013 and 20 May–10 December 2014. In practice, all the hilltop sample points were always measured during one day, while the rest of the sample points (or as many as possible) were measured during a 5-day period. In May–October, each sample point was measured on average every 22 days. Some of the sample points were measured significantly more often than others, each being measured 7–23 times during the two-year-campaign with a median of 13 measurements per sample point. The most active measurement period was June–August for both years.

**2.3 Flux calculation**

The procedure in flux calculations included: 1) filtering outliers from raw concentration data, 2) flux calculation using linear and non-linear functions, and estimating goodness-of-fit (GOF) parameters for the fluxes, 3) flagging the fluxes based on method quantification limit (MQL; Corley, 2003), 4) applying GOF criteria to flux data, and 5) creating final flux data.

We removed the outliers from the $CH_4$ mixing ratio data by using a robust regression analysis that uses iteratively reweighted least squares with a bisquare weighting function (Holland and Welsch, 1977), by setting a weight limit to 0.87 and discarded all points below this limit as outliers. The fluxes were calculated from the outlier-filtered raw data using both linear and exponential fit (for the calculation see Pihlatie et al. 2013). The exponential fit parameters were based on 17th order Taylor power series expansion (Kutzbach et al., 2007).

Firstly, decreasing $CO_2$ in opaque chamber or $CO_2$ flux below the MQL indicate a possible problem with the measurement, e.g. leaking chamber, and thus these measurements were omitted. For the $CH_4$ fluxes that were above MQL the following GOF criteria must be met for the flux to be included in the final flux data: normalized root mean square error (NRMSE) below 0.2 and the coefficient of determination ($R^2$) above 0.7. The fluxes below MQL were accepted in the final data as such, without applying the NRMSE and $R^2$ criteria, as neither of these GOF parameters work for close-to-zero fluxes. Furthermore, if the $CH_4$ mixing ratio was > 10 ppm in the beginning of the closure the flux was omitted. Finally, there was one exceptionally large $CH_4$ emission which was omitted from the final data set. The MQL of the GC was 0.10 ppm for $CH_4$ and 151 ppm for $CO_2$ (calculated according to Corley 2003). The $CH_4$ fluxes below MQL were between −3.74 and +2.38 $\mu mol\ m^{-2}\ h^{-1}$ for the larger chambers and between −0.146 and +0.244 $\mu mol\ m^{-2}\ h^{-1}$ for the smaller chambers, calculated by the linear fit. While in general linear fit tends to underestimate the chamber fluxes (Pihlatie et al., 2013), regarding small fluxes exponential fitting is more prone to errors and over-parameterization, and the relationship between linear and exponential flux values is more variable (Korkiakoski et al., 2017; Pedersen et al., 2010). Thus it is recommended to select between linear or non-linear fitting

depending on the concentration data (Korkiakoski et al., 2017; Pedersen et al., 2010). We used linear fit for all the fluxes that were below MQL.

After filtering the data, the final flux data included in total 723 measurements, of which 344 from year 2013 and 379 from year 2014. There were 5–21 measurements from each sample point, with a median of 11 measurements per point. In the final data set, 467 fluxes were calculated with exponential fit and 256 with linear fit, of which 184 were below the MQL.

**2.4 Environmental variables**

We measured soil moisture (volumetric water content) in A-horizon (ca. 5 cm depth) manually at the sample points (except for the hilltop area), simultaneously with the flux measurements (ThetaProbe ML2x with HH2 Moisture Meter, Delta-T Devices Ltd, Cambridge, UK). The soil moisture was calculated as an average of three recordings at a sample point. At the hilltop area, the soil moisture was measured continuously with a Time Domain Reflectometer (TDR-100, Campbell Scientific Inc., Utah, USA).

Soil temperature in A horizon was logged next to each sample point (except for the hilltop) eight times a day from June to October in 2013 and from April/May to October in 2014 with Thermochron iButton devices (Maxim Integrated Products, California, USA). At the hilltop area, the A horizon soil temperature was recorded automatically at five locations by silicon temperature sensors (KTY81-110, Philips, Netherlands) at 10-minute intervals. In the analysis, we used daily average soil temperatures of the flux measurement days at each sample point. For May–June in 2013, when the iButtons were not yet installed, we used the hilltop soil temperature data, as the temperature was rather consistent at all the sample points (average temperature in 12–30 June 2013 at the sample points ranged from 10.8 to 13.4 °C).

In addition to the continuous recordings of soil temperature and moisture, we used air temperature at 4.2 m height (Pt100 sensors with radiation shields by Metallityöpaja Toivo Pohja), and precipitation (Vaisala FD12P weather sensor at 18 m height) measured at the SMEAR II station.

**2.5 Ground vegetation of the sample points**

The composition of ground vegetation in 54 sample points (all except the hilltop points) was described by estimating projection cover of each plant species with the help of a frame divided into 0.1 x 0.1 m sectors. Cover less than 5% was marked to be 3%. To group the sample points based on their plant composition we performed a Two-way indicator species analysis (TWINSPAN), a divisive clustering method using TWINSPAN for Windows version 2.3 (Hill and Šmilauer, 2005). We used moss species as indicators because their distribution is generally more strongly related to soil moisture than the distribution of clonal vascular plants typical to boreal forests (e.g. Hokkanen, 2006). For a robust result we excluded species which frequency was less than three. Based on the plant species composition in 54 sample points we created four vegetation groups. To visualize the variation within and between the groups we performed Canonical Correspondence Analysis (CCA) where vegetation groups from TWINSPAN were used as environmental variables. Before CCA Detrended Correspondence Analysis (DCA) was conducted to decide between linear and unimodal methods. Canoco 5.11 (Šmilauer and Lepš, 2014) was applied for both analyses.

**2.6 Statistical analyses**

Multiple-group comparisons were performed with one-way ANOVA, and two-group-comparisons with t-test, when the Levene's test indicated equal variances, and distribution was normal or sample size large enough. When groups had unequal variances, we used Welch's ANOVA (Analysis of Variance) together with Games-Howell test as a post hoc test. When comparing two groups with unequal variances, we used Welch's t-test, or Satterthwaite's approximation. When groups had equal variances, but distribution was non-normal, or sample size was very small, we used Kruskal-Wallis, followed by Bonferroni correction for pairwise comparisons.

The Spearman's correlation coefficients were calculated to study the relationships between the $CH_4$ fluxes and the environmental / topographical parameters at the camber locations. The Spearman's correlation was also performed between the $CH_4$ flux, soil moisture, and soil temperature time series data, as the correlations were not linear. Soil temperature can increase both $CH_4$ emissions and uptake, via increasing the activity of the soil microbes, and thus we used absolute flux values when examining the effect of the temperature.

Welch's t-test, Welch's ANOVA and Kruskal-Wallis, accompanied by the post hoc tests, were performed with SPSS (IBM SPSS Statistics 24, New York, USA). Regular one-way ANOVA, Levene's tests, and correlation analyses were performed with MATLAB (R2017b / R2018b, MathWorks, Natick, Massachusetts, USA). The statistical analyses were assessed at a significance level of $p<0.05$.

**2.7 Spatial drivers of the $CH_4$ flux**

In order to find the spatial parameters connected to the $CH_4$ flux, we obtained the following spatial data sets for the study site: digital elevation model (DEM) (Elevation model derived from airborne lidar scanning, National Land Survey of Finland, 2/2019), biomass of foliage (for pine, spruce, and broadleaf trees) and tree volumes (for pine, spruce, and birch) (16x16 m; Multi-source National Forest Inventory, The Natural Resources Institute Finland, 2015; Mäkisara, Katila, & Peräsaari, 2019), subsoil types (basal deposit at a depth of 1 m) (16x16 m; Superficial deposits 1:20 000/1:50 000, Geological Survey of Finland, 2015), and peated soil areas (16x16 m; Topographic database, National Land Survey of Finland, 8/2018). In addition, we calculated the following topographic indices from the DEM: topographic wetness index (TWI; Beven and Kirkby, 1979), terrain ruggedness index (TRI; Riley et al., 1999), slope, and cartographic depth-to-water index (DTW; Murphy et al., 2007) (Appendix A, Figs. A1–4). Before calculating the indices, the DEM was pre-processed to be hydrologically correct by using TopoToolbox 'carve' option (Schwanghart and Kuhn, 2010). TWI was calculated as a natural logarithm of the ratio between the specific catchment area (contributing area per unit contour length) and tangent of the local slope. The upslope catchment area was calculated using multiple flow direction algorithm (Freeman, 1991; Schwanghart and Kuhn, 2010), and local slope was calculated using adjacent points in DEM. The calculations were made with TopoToolbox. Following the recommendations by Ågren et al., (2014), TWI was calculated from coarse resolution (16 m) DEM, and scaled back to a finer 5 m grid with bilinear interpolation, since Ågren et al. (2014) found that TWI calculated from coarse grid represented the soil moisture better than when calculated from a finer grid. The other indices were calculated from DEM with 5 m resolution. TRI was calculated using the gdaldem program which is part of Geospatial Data Abstraction Library (GDAL). TRI describes the amount of elevation difference between adjacent cells in a DEM grid, and hence presumably has a low value on flat hill tops and depressions. The flow channel networks in the study domain used for the DTW calculations were estimated from the DEM using one-hectare flow initiation threshold (Ågren et al., 2014), and then the DTW values were calculated using r.cost function of GRASS GIS where the terrain

slope raster map was used as a cost layer (e.g. Murphy et al., 2007). DTW can be considered to describe the elevation difference to the nearest open water location derived from the DEM.

**2.8 Modelling the soil moisture and the CH₄ flux to the site**

We used Random Forest (RF) algorithm (Breiman, 2001) to upscale the soil moisture and $CH_4$ flux to the whole area for two time periods: May–July and August–October. The primary purpose of this upscaling was to get an accurate estimate of landscape-level forest floor $CH_4$ fluxes in a way that reflects the soil heterogeneity. We opted to do two static predictions for two separate periods instead of trying to capture the temporal variability, as we did not have enough temporal data from each sample point, and modelling temporal variability of soil moisture has been shown to be difficult even with larger data sets than the one used here (Kemppinen et al., 2018). All the 60 sample points, for which data were temporally averaged for the two separate periods (on average 7 and 5 measurements for each point during May–July and August–October, respectively), were used in RF model development. RF is a machine learning algorithm that can be used to generalise complex dependencies between driving variables and a target variable. Here, our RF models consisted of a large ensemble of regression trees, which were trained each with a separate random subsample of available data. The output from the RF model is an average of output from all the trees separately – and hence the algorithm applies the bootstrap aggregation (bagging) method, which decreases the noise of the prediction. The model for soil moisture was developed using four drivers (TWI, slope, DTW and TRI), and for $CH_4$ flux using five drivers (soil moisture, TWI, slope, DTW and TRI), selected based on the correlations (Appendix A, Table A1, Figs. A5–6). MATLAB (R2018b, MathWorks, Natick, Massachusetts, USA) function TreeBagger was used for developing the RF models in this study. Each trained forest consisted of 300 regression trees and minimum of two samples were allowed in a split node. Values for these hyperparameters were selected based on initial testing using out-of-bag errors and the minimum number of observations per tree leaf was set to two, due to limited amount of data available. However, the number of variables randomly sampled as candidates at each split was not changed from its default value (one third of the total number of variables).

The predictive performance of the developed RF models were evaluated using distance-blocked leave-one-out cross validation. In this method, one RF model is developed for each sample point location, and the training data consist of data measured further than selected distance (here 30 meters) from the sample point location in question, while the rest of the data (i.e. data originating from closer than 30 meters) are utilised as independent validation data. This way the possible spatial autocorrelations in the data did not inflate the cross-validation metrics (Appendix A, Fig. A7–9). Blocked cross-validation has been proposed to be appropriate cross-validation strategy for data showing e.g. spatial autocorrelations (Roberts et al., 2017), and it has been used in some prior flux upscaling exercises (Peltola et al., 2019). Statistical metrics used in evaluating model predictive performance included mean bias, fraction of variance explained by the model ($R^2$), Nash-Sutcliffe model efficiency (NSE) and root mean squared error (RMSE). The uncertainty of the upscaled soil moisture and $CH_4$ flux, however, was estimated by developing 100 RF models with a random subset (70%) of available data, and the variability (standard deviation) over this ensemble was used as an uncertainty estimate. The uncertainty estimate describes the robustness of the soil moisture and $CH_4$ flux dependence on the drivers identified by the RF model, but likely also the effect of the distribution of the sampling points. This approach is similar to the one used in e.g. Peltola et al. (2019) and Aalto et al., (2018).

After modelling the soil moisture to the whole study domain, the forest floor $CH_4$ flux was modelled as well using the produced soil moisture raster map along with the maps of topographic metrics used to develop the RF model for $CH_4$ flux.

## 2.9 Evaluating the representability of chamber measurements

The representability of $CH_4$ flux chamber measurements was evaluated by comparing the average of $CH_4$ fluxes measured at $n$ chamber locations against the mean $CH_4$ flux modelled for the whole study domain with the RF modelling approach (Sect. 2.8). The aim of this analysis was to evaluate how many chamber measurement locations are needed to get an accurate estimate of landscape-level flux by only averaging over the measured chamber data without any upscaling with e.g. RF algorithms. The mean RF-modelled $CH_4$ flux is used as a reference as it accounts for the soil heterogeneity. The $CH_4$ fluxes measured at $n$ chamber locations were selected by random and averaged. This was repeated for maximum 500 times for each $n$ and mean absolute bias between these estimates, and the reference was calculated as a metric for the representability of $n$ chamber locations.

## 3 Results

### 3.1 Ground vegetation at the sample points

The most common vascular species growing in the sample points were *V. vitis-idaea* (48 out of 54 studied sample points), *V. myrtillus* (45 points), *Equisetum sylvaticum*, and *L. borealis*, followed by *M. bifolium* and *T. europaea*. The most prevalent mosses were *P. schreberi* (34 points), *Polytrichum commune* (29 points), *Sphagnum* spp. (28 points), *D. polysetum*, and *H. splendens*.

The four vegetation groups were named based on their dominant mosses. (1) Sphagnum-group included 15 sample points that had over 50% *Sphagnum* coverage (except for one point) and no *P. schreberi*. Distinctive species were also *E. sylvaticum*, *Carex digitata*, and *P. commune* that all are typical to peatland forest. (2) Sphagnum-Pleurozium-group is an intermediary group between the swampy and drier forest areas with some *Sphagnum* but also *P. schreberi* in all its 13 sample points. Sample points in the remaining two groups do not include any *Sphagnum* but species characteristic to upland forests. Sample points in (3) Pleurozium-group have typically more than 10% *P. schreberi*, while the sample points in (4) Hylocomium-group have less *P. schreberi* and usually more than 80% *H. splendens*, which is related to slightly higher fertility. *D. polysetum* was most common in Pleurozium-group, and some of the Pleurozium-group had high *L. borealis* coverage. *V. vitis-idaea* and *V. myrtillus* were common in all the groups – their coverage was the highest in Pleurozium-group and the lowest in Sphagnum group. Finally, the hilltop sample points were put to the Pleurozium-group based on their vegetation. The Pleurozium-group has thus in total 25 sample points, and the Hylocomium-group has seven.

### 3.2 Soil moisture and temperature at the site

The annual precipitation at the site was 576 mm in 2013 and 572 mm in 2014. The annual mean air temperature was 5.0 °C in 2013 and 5.2 °C in 2014. The mean soil temperature at the top of the hill was 5.9 °C in 2013 and 6.0 °C in 2014, and the mean soil moisture was 0.25 $m^3$ $m^{-3}$ in 2013 and 0.24 $m^3$ $m^{-3}$ in 2014. The mean air temperature of May–October varied between 10.0–13.2 °C in 2010–2017, being 12.4 °C in 2013 and 12.7 °C in 2014. The years 2013 and 2014 were somewhat warmer and had less precipitation compared to the long-time averages reported in

Pirinen et al. (2012), and the annual precipitation values (576 and 572 mm in 2013 and 2014, respectively) were lower compared to the adjacent years (2010–2012 and 2015–2017; 678–925 mm) at the measurement site. The soil moisture as well as the soil temperature and precipitation followed a similar temporal pattern in both measurement years, although the spring was a bit wetter in 2013 (Fig. 2). This difference was largely due to thicker snow cover and later snowmelt in 2013 (mean snow cover in December–February 37 cm; snowmelt in late April) compared to 2014 (mean snow cover in December–February 7 cm; snowmelt in mid-March). The measurement years were rather similar regarding the weather conditions, which allowed us to combine the measured data from two years.

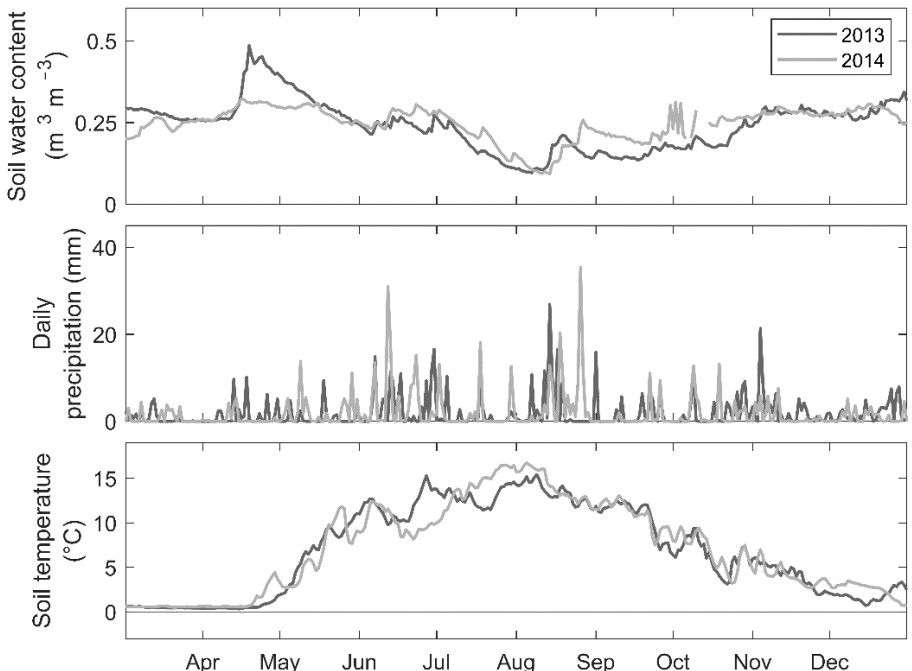

**Figure 2. Daily mean soil moisture (volumetric water content, m³ m⁻³) in A horizon, daily precipitation (mm), and daily mean soil temperature (°C) in A horizon, between March–December in 2013 and 2014 at the SMEAR II station (on top of the hill).**

The mean soil moisture of the sample points was ranging from 0.09 (±0.03 SD) m³ m⁻³ (sample point NE–E-3) to 0.89 (±0.12 SD) m³ m⁻³ (E–SE-6). Furthermore, there was a significant difference in the mean soil moisture between the seasons May–July and August–October (0.25 (±0.06 SD) and 0.18 (±0.05 SD) m³ m⁻³, respectively, SMEAR II continuous measurements, or 0.35 (±0.28 SD) and 0.29 (±0.22 SD) m³ m⁻³, respectively, at the sample points on measurement days; $p<0.0001$). The mean soil moisture of the sample points differed between the two subsoil types (sandy/gravelly till 0.37 (±0.24 SD) m³ m⁻³, bedrock with shallow moraine layer 0.26 (±0.24 SD) m³ m⁻³; $p<0.001$), while there was no significant difference in the soil temperature.

The continuous measurements of the SMEAR II station show that the soil temperature in A horizon was between 0.3 and 15.4 °C in 2013, and between –1.8 and 16.8 °C in 2014 (January–December) (Fig. 2). There was a steep increase in the soil temperature at the turn of April and May in both years, peaking in July–August (Fig. 2). Between May–October the soil temperature was 1.9–15.4 °C in 2013, and 2.7–16.8 °C in 2014. There was no significant difference between the years. Soil temperature was not as spatially variable as soil moisture (no statistically significant differences between sample points). The average soil temperature of the measurement days at the sample points was between 9.8 (±4.1 SD) °C (H6) and 13.1 (±1.8 SD) °C (W–NW-4).

### 3.3 Forest floor CH₄ flux

The mean measured CH₄ flux at the site in 2013–2014 was −4.18 (± 43.2 SD) µmol m⁻² h⁻¹, and the median was −6.07 µmol m⁻² h⁻¹ ($n$=723). The measured fluxes ranged from −56.8 to 1080 µmol m⁻² h⁻¹.

Emissions of CH₄ were measured in total 63 times from 17 different sample points, corresponding to 9% of the measurements and 28% of the points. Most of the CH₄-emitting sample points belonged to the Sphagnum-group (11 out of 17), and the highest CH₄ emissions were detected from six sample points in the Sphagnum-group (sample

points SW–W-2–3, E–SE-4–6, and S–SW-4). No emissions were observed from the Hylocomium-group sample points.

The highest CH₄ emission was detected on 5 June 2013 from SW–W-3 (Spaghnum-group), which was located at a water-filled patch (water table was above the peat surface for most of the time). The CH₄ flux from this sample point ranged between −33.4 and 1080 µmol m⁻² h⁻¹, the mean being 107 µmol m⁻² h⁻¹. The highest emission is excluded

from all further statistical analyses as an outlier, nevertheless, there was no indication of fault in the measurement. Consequently, without the highest emission, the mean CH₄ flux from all the sample points at the site in 2013–2014 was −5.69 (±14.9 SD) µmol m⁻² h⁻¹, and the maximum was 212 µmol m⁻² h⁻¹ ($n$=722).

There was a significant positive correlation between the measured soil moisture and the CH₄ fluxes ($r_s$ = 0.70, $p$<0.001; $n$=722). If we explore the years separately, the relationship was slightly stronger in 2014 than in 2013 ($r_s$ =

0.73 and $r_s$ = 0.62, respectively, $p$<0.001; $n$=722). There was also a statistically significant positive correlation between the mean CH₄ flux and the mean soil moisture at the sample points ($r_s$ = 0.78, $p$<0.001; $n$=60) (Fig. 3). The absolute values of the CH₄ flux and the daily mean soil temperature at the sample points had a weak positive correlation ($r_s$ = 0.22, $p$<0.001; $n$=722), and the correlation was similar in early summer and late summer.

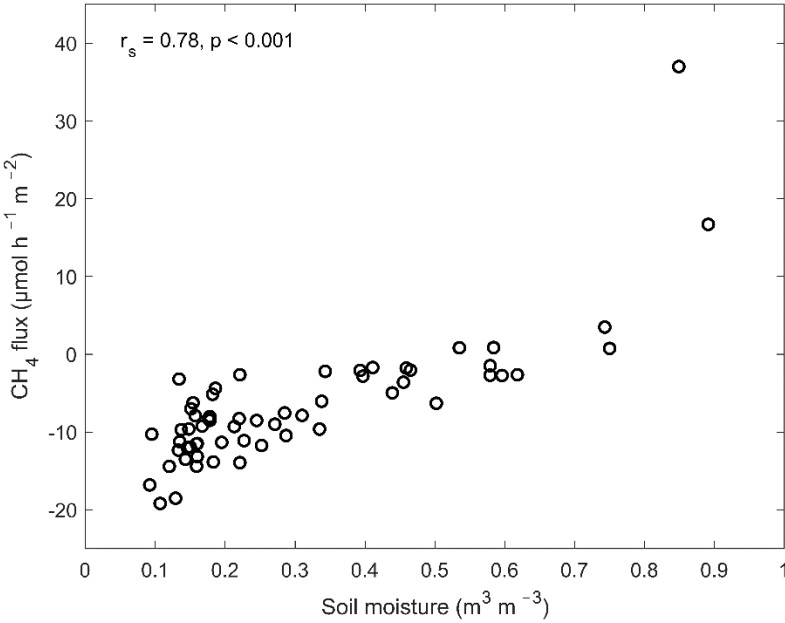

**Figure 3. The relationship between the mean of the measured soil moisture (volumetric water content, m³ m⁻³) and the mean of the measured CH₄ flux (µmol h⁻¹ m⁻²) at the sample points (Spearman's correlation coefficient, $r_s$ = 0.78, $p$<0.001).**

The mean measured CH₄ flux in May–October at the site was −4.88 (±20.3 SD) µmol m⁻² h⁻¹ (median −6.43 µmol m⁻² h⁻¹) in 2013 ($n$=339), and −6.46 (±7.47 SD) µmol m⁻² h⁻¹ (median −5.90 µmol m⁻² h⁻¹) in 2014 ($n$=373), however, the difference was not statistically significant. Furthermore, the mean CH₄ flux differed between the early summer

and autumn seasons (May–July −3.49 (±19.0 SD; $n$=392) µmol m$^{-2}$ h$^{-1}$ and August–October −8.42 (±6.79 SD; $n$=320) µmol m$^{-2}$ h$^{-1}$; $p$<0.0001). The CH$_4$ flux was slightly more dependent on soil moisture in May–July than August–October ($r_s$ 0.75 and 0.62, respectively; $p$<0.0001).

There were significant differences between the vegetation groups in both the soil moisture and the CH$_4$ flux, in both seasons (Fig. 4). The mean soil moisture was decreasing from the Sphagnum-group to the Pleurozium-group, and the

380 differences between the groups were statistically significant, except between the two driest groups (Fig. 4a, b). The mean CH$_4$ flux decreased from the Sphagnum-group to the Hylocomium-group, indicating mean CH$_4$ emission from the Sphagnum-group sample points in May–July (Fig. 4c), and increasing CH$_4$ uptake from the Sphagnum-Pleurozium-group to the Hylocomium-group ($p$<0.05) (Fig. 4c, d).

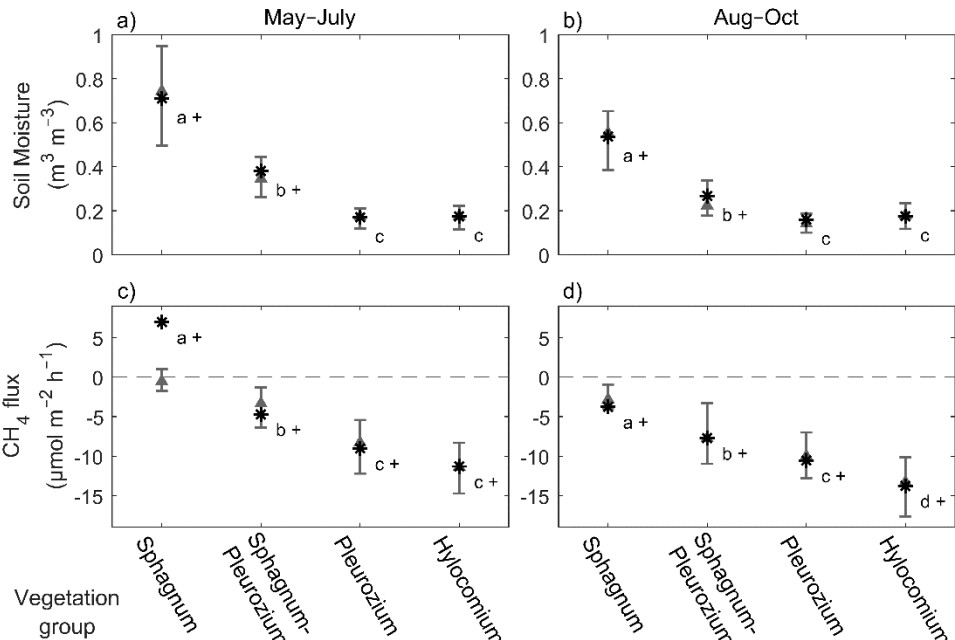

**Figure 4. The soil moisture (volumetric water content, m$^3$ m$^{-3}$) (a, b) and the measured CH$_4$ flux (µmol m$^{-2}$ h$^{-1}$) (c, d) of the different vegetation groups in May–July and August–October. The asterisks indicate the mean values, triangles indicate the medians, and the whiskers show the 25th and 75th percentiles. Statistically significant differences ($p$<0.05) within each subplot are marked with different letters, and plus signs indicate significant differences between the seasons in a group. Sample points in the Sphagnum group have >50% cover of *Sphagnum* sp., while Hylocomium group sample**
**points have typically ≥80% of *Hylocomium splendens*, and the others are intermediary groups between the extremes.**

The mean soil moisture decreased between the two seasons in Sphagnum-group (May–July 0.71 (±0.24 SD) m$^3$ m$^{-3}$, August–October 0.54 (±0.23 SD) m$^3$ m$^{-3}$; $p$<0.0001) and Sphagnum-Pleurozium-group (May–July 0.38 (±0.17 SD) m$^3$ m$^{-3}$, August–October 0.27 (±0.14 SD) m$^3$ m$^{-3}$; $p$<0.001), but remained the same in the rest of the groups (0.16–0.17 (±0.07–0.09 SD) m$^3$ m$^{-3}$) (Figs. 4a, b). The mean CH$_4$ flux in the Sphagnum-group turned from emission (6.96

±31.2 µmol m$^{-2}$ h$^{-1}$) to uptake (−3.76 ±5.14 µmol m$^{-2}$ h$^{-1}$) ($p$<0.001), and the uptake increased significantly in the rest of the vegetation groups between early summer and late summer ($p$<0.05).

When considering the spatial variation of the CH$_4$ flux at the site, the measured CH$_4$ fluxes differed markedly between the locations (groups of 2–6 sample points) (Fig. 5). Two sample point groups had a mean flux indicating CH$_4$ emissions: SW–W-1–3 (9.45 ±35.8 SD µmol m$^{-2}$ h$^{-1}$) and E–SE-4–6 (7.18 ±32.9 µmol m$^{-2}$ h$^{-1}$) (Fig. 5). However,

all the median fluxes of the sample point groups were negative. Emissions were measured from nine groups. The

strongest mean $CH_4$ uptake was measured from group SE–S–4–6 (–14.26 ±10.4 µmol m$^{-2}$ h$^{-1}$). The mean $CH_4$ flux of all the six hilltop points was –8.63 ±4.11 µmol m$^{-2}$ h$^{-1}$ (Fig. 5). Between May–July the hilltop mean flux was –8.01 ±3.22 µmol m$^{-2}$ h$^{-1}$, and in August–October –10.3 ±5.04 µmol m$^{-2}$ h$^{-1}$.

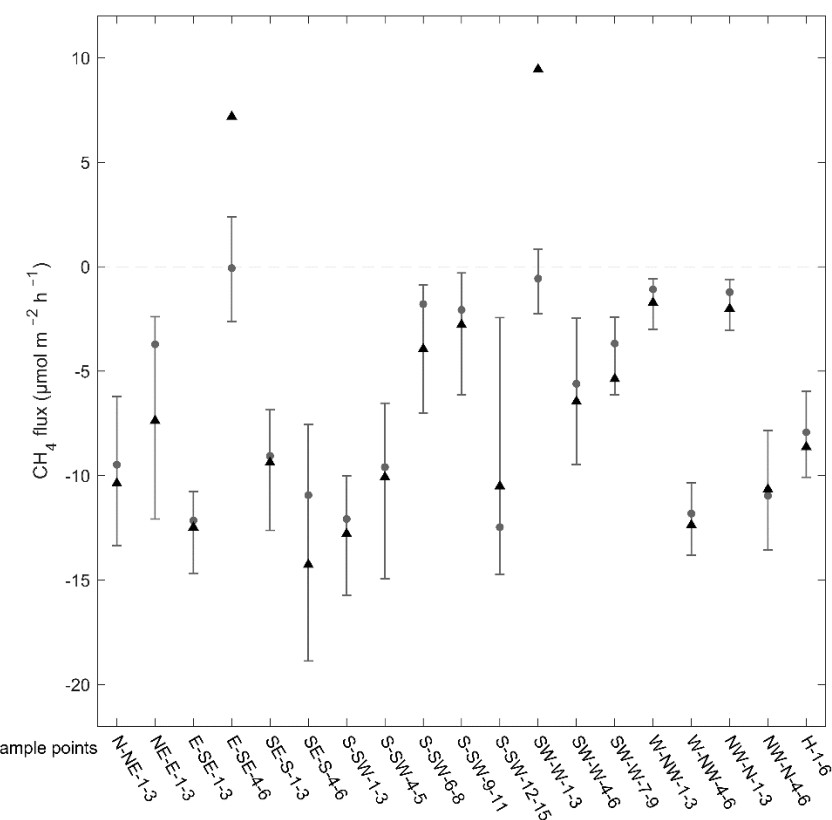

**Figure 5. Fluxes of CH$_4$ (µmol CH$_4$ m$^{-2}$ h$^{-1}$) from the sample point groups. Mean values are represented with triangles, medians with circles, and the whiskers show 25$^{th}$ and 75$^{th}$ percentiles.**

### 3.4 Modelled soil moisture and CH$_4$ flux at the site

The performance of the RF model to predict soil moisture variability in the study domain is outlined with the statistical metrics shown in Table 1 (see also Appendix Fig. A8). These metrics were calculated against independent validation data and hence are suitable for assessing the predictive performance of the model. $R^2$ values were 0.51 and 0.26 for May–July and August–October, indicating that the model was able to describe the spatial variation in soil moisture more accurately during the early summer period. The RMSE and mean bias in the model prediction were similar during both periods.

**Table 1. Statistical metrics outlining the predictive performance of the RF model for the soil moisture during the two seasons. The metrics were calculated using distance-blocked cross-validation technique.**

|                | $R^2$ | NSE   | RMSE (m$^3$ m$^{-3}$) | Bias (m$^3$ m$^{-3}$) |
|----------------|-------|-------|-----------------------|-----------------------|
| May–July       | 0.51  | 0.18  | 0.17                  | 0.014                 |
| August–October | 0.26  | −0.25 | 0.17                  | 0.015                 |

The modelled soil moisture of the whole studied area ranged in May–July from 0.11 to 0.79 $m^3$ $m^{-3}$, and in August–October from 0.12 to 0.65 $m^3$ $m^{-3}$. The mean soil moisture in May–July was 0.25 $m^3$ $m^{-3}$ (± 0.12 SD) and in August–October 0.23 $m^3$ $m^{-3}$ (± 0.09 SD) (Table 2). The mean of the modelled values at the sample point locations (0.33 and 0.29 $m^3$ $m^{-3}$ in May–July and August–October, respectively) were similar to the measured mean values (0.33 and 0.28 $m^3$ $m^{-3}$ in May–July and August–October, respectively), while the modelled averages for the whole area were slightly lower for both seasons (Table 2). Based on the RF-modelled soil moisture, the site was mostly rather dry in 2013–2014, with some wetter areas where the soil moisture was above 0.5 $m^3$ $m^{-3}$ (Fig. 6). The modelling results indicated that the wet areas were wetter and wider in May–July (Fig. 6a) than in August–October (Fig. 6b), which follows from the measured soil moisture (see Sect. 3.2). In May–July 5% of the area was wet (soil moisture >0.5 $m^3$ $m^{-3}$), whereas in August–October only ca. 1% of the area could be considered wet. The RF model predicted high soil moisture for topographical depressions and flat areas, which were specified by low DTW, slope and TRI, and high TWI values. The soil moisture was spatially more variable in early summer than in autumn. The relative uncertainty of the upscaled soil moisture was on average 12% and 9.6% during May–July and August–October, respectively (Fig. 7). The uncertainty increased with the predicted soil moisture, yet during May–July the wettest locations (soil moisture above 0.65 $m^3$ $m^{-3}$) had on average smaller relative uncertainty than the locations with intermediate (soil moisture between 0.4 and 0.65 $m^3$ $m^{-3}$) wetness (5.5% and 9.7%, respectively). This indicates that the RF model was able to better constrain the moisture variability at wet depressions and at very dry locations than at areas with intermediate wetness, using the drivers used to develop the model (see Sect. 2.7).

**Table 2. The means (± standard deviation) and ranges of the measured and modelled CH$_4$ fluxes (µmol m$^{-2}$ h$^{-1}$) and the measured and modelled soil moisture (m$^3$ m$^{-3}$) for the two seasons (May–July and August–October).**

| | Measured (sample points) | | | Modelled (sample points) | | | Modelled (whole area) | | |
|---|---|---|---|---|---|---|---|---|---|
| | Mean | SD | Range of sample point means | Mean | SD | Range | Mean | SD | Range |
| **CH$_4$ flux (µmol m$^{-2}$ h$^{-1}$)** | | | | | | | | | |
| May–July | −5.07 | (±11.0) | −20.2 to 58.5 | −5.84 | (±4.67) | −11.6 to 6.19 | −7.42 | (±3.26) | −12.3 to 6.19 |
| August–October | −8.67 | (±5.12) | −24.1 to −1.31 | −8.51 | (±3.80) | −13.0 to −2.12 | −9.91 | (±2.73) | −14.6 to −2.12 |
| **Soil Moisture (m$^3$ m$^{-3}$)** | | | | | | | | | |
| May–July | 0.33 | (±0.24) | 0.066 to 0.92 | 0.33 | (±0.20) | 0.11 to 0.77 | 0.25 | (±0.12) | 0.11 to 0.79 |
| August–October | 0.28 | (±0.18) | 0.089 to 0.86 | 0.29 | (±0.16) | 0.12 to 0.65 | 0.23 | (±0.091) | 0.12 to 0.65 |

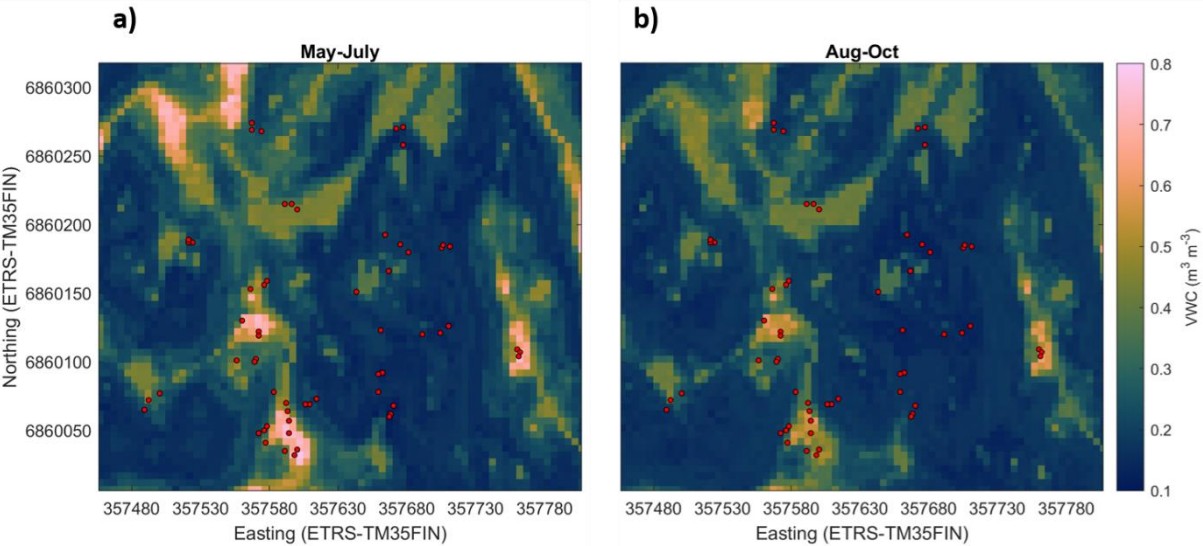

**Figure 6. Modelled soil moisture (volumetric water content, m³ m⁻³) at the site in May–July (a) and August–October (b). The red circles show the sample points.**

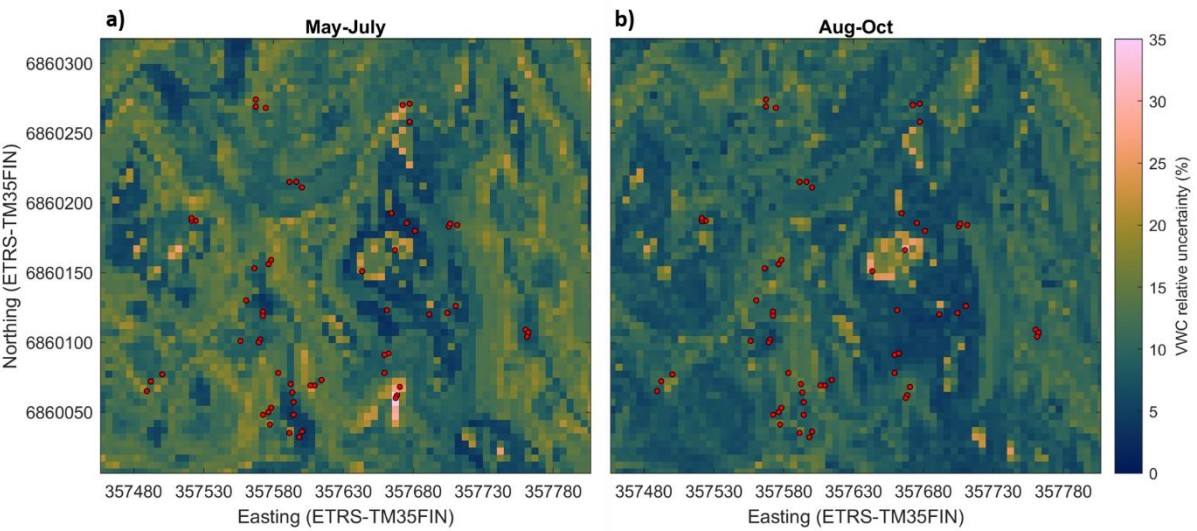

**Figure 7. Relative uncertainty of the modelled soil moisture (volumetric water content, %) at the site in May–July (a) and August–October (b). The relative uncertainty was defined as the standard deviation of the results of individual Random Forest models in the RF model ensemble divided by the modelled soil moisture (see Sect. 2.8). The red circles show the sample points.**

Based on cross-validation, the agreement between the upscaled and the measured CH₄ fluxes at the different chamber

locations was moderate ($R^2$=0.26 and $R^2$=0.39 for May–July and August–October, respectively) (Table 3; see also Appendix Fig. A9). Note that the upscaling had difficulties especially in representing the tails of the CH₄ flux distribution (strong uptake/emission) which has an influence on the cross-validation metrics. The mean bias (upscaled−measured) was 0.10 and −0.10 µmol m⁻² h⁻¹ for the May–July and August–October periods, respectively. The modelled CH₄ flux of the whole studied area was between −12.3 and 6.17 µmol m⁻² h⁻¹ in May–July, with a

mean flux of −7.42 µmol m⁻² h⁻¹ (± 3.26 SD) (Table 2). In August–October, the flux ranged from −14.6 to –2.12 µmol m⁻² h⁻¹, with a mean of −9.91 µmol m⁻² h⁻¹ (± 2.73 SD) (Table 2). The modelled fluxes resulted in stronger CH₄ uptake than averaging the flux measurements of 60 sample points (Table 2). The upscaling demonstrated some

CH$_4$ emitting patches in the early summer (Fig. 8a), which shifted to CH$_4$ uptake in the autumn (Fig. 8b). The emission patches covered approximately 3% of the study area, and the flux of the emitting areas was 0.029–6.19 µmol m$^{-2}$ h$^{-1}$
in May–July. Omitting the emission patches from calculation would result in ca. 4% stronger mean CH$_4$ uptake in May–July. The soil moisture of the emitting cells was between 0.39–0.79 m$^3$ m$^{-3}$ in May–July, with a mean of 0.60 m$^3$ m$^{-3}$. In autumn the emission patches were drier, the soil moisture of these areas being 0.38–0.65 m$^3$ m$^{-3}$ with a mean of 0.48 m$^3$ m$^{-3}$ (±0.065 SD), and the CH$_4$ flux was between −5.31 and −2.12 µmol m$^{-2}$ h$^{-1}$ with a mean of −3.67 µmol m$^{-2}$ h$^{-1}$ (±0.615 SD). The relative uncertainties of the upscaled forest floor CH$_4$ fluxes were larger at the wet
depressions than at dry areas (Fig. 9), however the absolute uncertainties were lower. The upscaled CH$_4$ flux uncertainty showed similar spatial pattern during both periods. The lower absolute uncertainty at wet depressions was due to lower variability of the measured CH$_4$ fluxes between measurement locations at wet spots. Note that these uncertainty maps represent only the uncertainty stemming from the upscaling procedure with the RF model. Cross-validation metrics presented above are better for evaluating the overall uncertainty, which includes e.g. uncertainties
related to possibly biased sampling locations.

**Table 3. Statistical metrics outlining the agreement between upscaled and measured CH$_4$ fluxes at different chamber locations.**

|  | R$^2$ | NSE | RMSE (µmol m$^{-2}$ h$^{-1}$) | Bias (µmol m$^{-2}$ h$^{-1}$) |
|---|---|---|---|---|
| May–July | 0.26 | -0.96 | 9.5 | 0.1 |
| August–October | 0.39 | -0.50 | 4.0 | -0.1 |

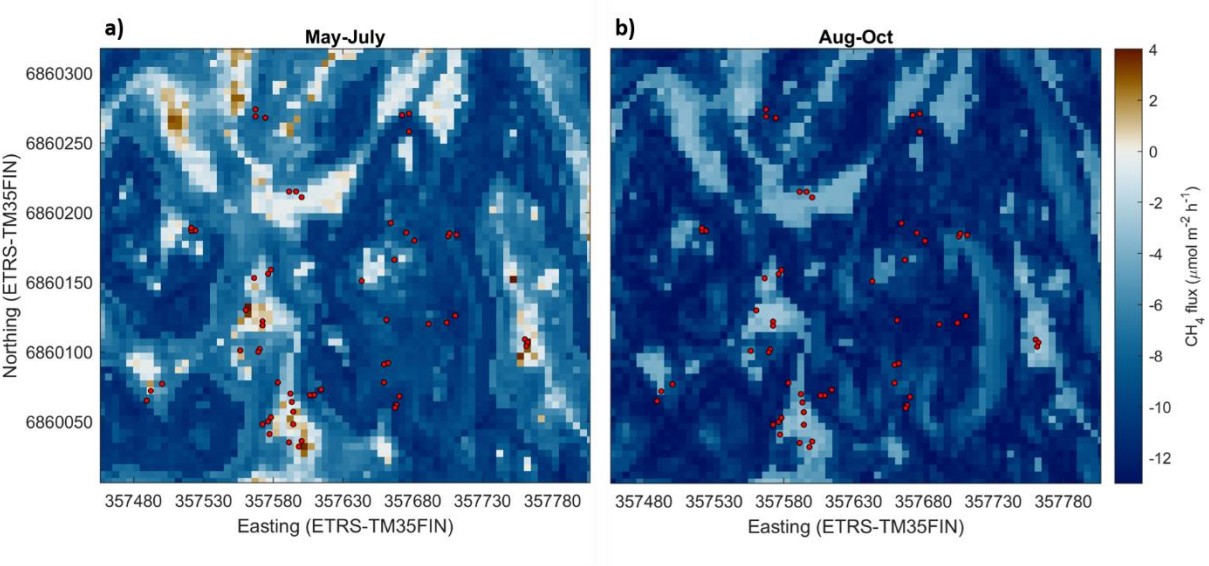

**Figure 8. Modelled forest floor CH$_4$ flux (µmol m$^{-2}$ h$^{-1}$) at the site in May–July (a) and August–October (b). (Values below zero indicate uptake and values above zero emission). The red circles show the sample points.**

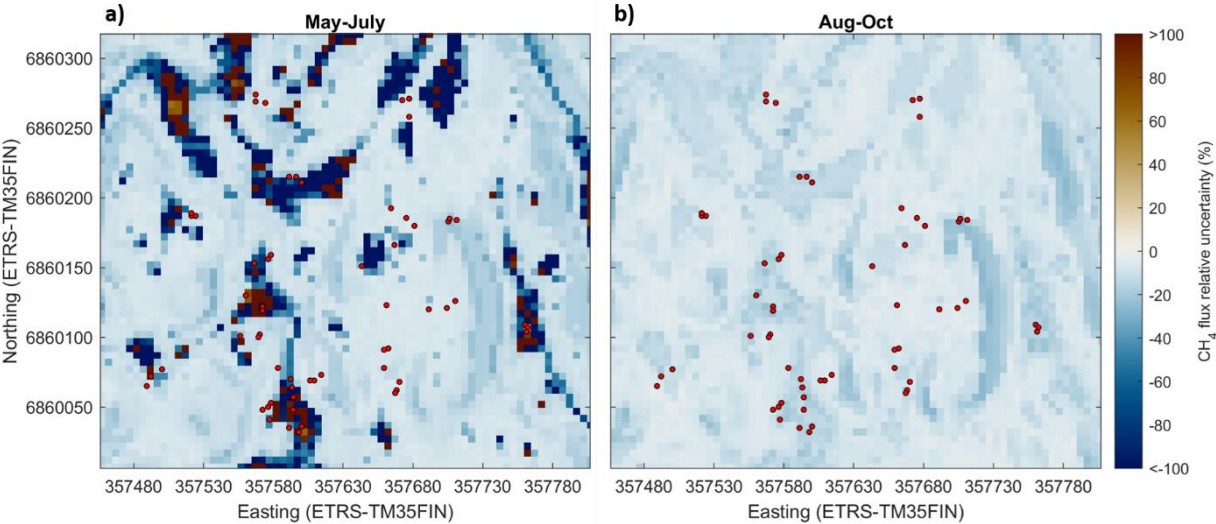

**Figure 9. Relative uncertainty of the modelled forest floor CH$_4$ flux (%) at the site in May–July (a) and August–October (b). The relative uncertainty was defined as the standard deviation of the results of individual Random Forest models in the RF model ensemble divided by the modelled CH$_4$ flux (see Sect. 2.8). The red circles show the sample points.**

In order to evaluate the number of sample points needed to produce accurate estimate of the landscape-level forest floor CH$_4$ flux, we compared mean CH$_4$ flux from 1–60 randomly picked sample points with the mean of the upscaled flux (Sect. 2.9). Based on this comparison, we state that with approximately 15–20 randomly selected sample points similar accuracy was achieved as averaging over all the sample points (Fig. 10). This finding held irrespective of the study period investigated.

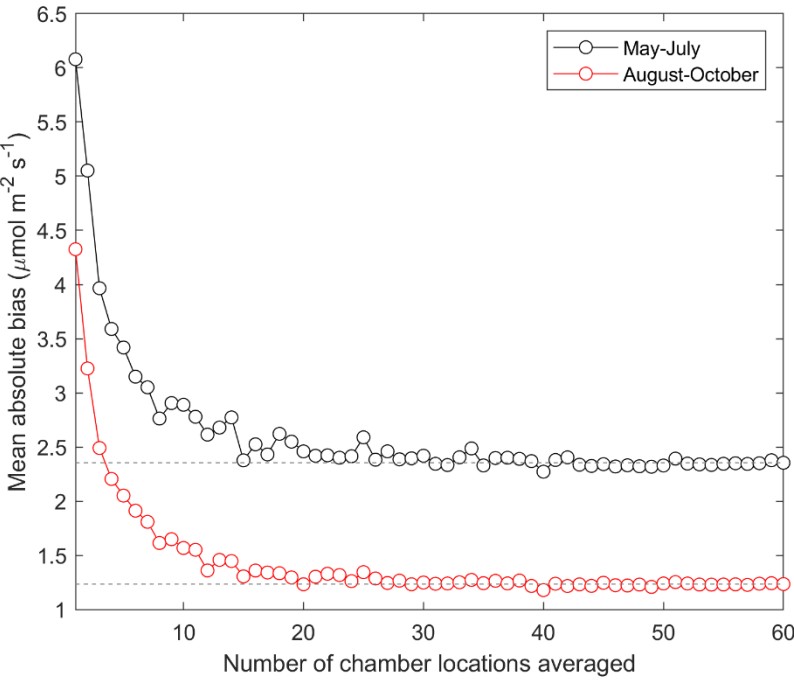

**Figure 10. Mean absolute bias in the measured forest floor CH$_4$ flux estimated from a random subset of the sample points. Mean upscaled CH$_4$ flux was used as a reference. (Max. 500 sample point combinations were calculated.)**

## 4 Discussion

### 4.1 Drivers and spatial variation of the CH$_4$ flux in the two seasons

We performed spatially extensive CH$_4$ flux measurements from the forest floor, covering different soil moisture conditions and vegetation types at the area, and combined the measured data with remote sensing tools in order to unravel the spatial variation in the forest floor CH$_4$ flux within the terrain. We accomplished this by generating random forest (RF) models to map the soil moisture and the CH$_4$ flux at the boreal forest site. Our results demonstrate that even though the forest floor is mainly a sink of CH$_4$, as expected for the mainly dry upland pine forest, the CH$_4$ flux at the site is highly heterogeneous.

In our study, the soil moisture was the most important driving force in the spatial variability of the CH$_4$ fluxes, whereas soil moisture is highly driven by topography. In previous studies, soil moisture and TWI have also been identified as the main factors affecting the soil CH$_4$ fluxes on a spatial perspective (Kaiser et al., 2018; Warner et al., 2019). Furthermore, soil moisture and vegetation are strongly interconnected: while topography-driven soil moisture controls vegetation (Moeslund et al., 2013), vegetation can also affect soil moisture via e.g. evapotranspiration (Dunn and Mackay, 1995) or rooting strategies (Milly, 1997). As the soil moisture is affected by topography, vegetation, and soil properties, it can have high spatial variability (Rosenbaum et al., 2012). Based on our results, there was high variation in the CH$_4$ fluxes even within the sample point groups, which may be explained by different vegetation groups among adjacent sample points.

Our vegetation classification was mostly based on mosses, and the connection to soil moisture was expected. Consequently, the groups differed in their soil moisture, and furthermore, in their CH$_4$ flux. However, while the soil moisture did not differ between the Pleurozium-group and the Hylocomium-group, there was a significant difference in the CH$_4$ fluxes in autumn, suggesting that the ground vegetation as such may affect the CH$_4$ flux. Some plant species of forest ground vegetation have been suggested to contribute to CH$_4$ exchange and the effect may vary between species (Halmeenmäki et al., 2017; Maier et al., 2017; Praeg et al., 2017), yet the studies focusing on forest ground vegetation are still rare. Also different tree species can affect the soil conditions and thus the forest floor CH$_4$ flux, for example through soil chemical properties or nutrient conditions (Reay et al., 2005). While soil moisture undergoes also short-term temporal changes resulting from weather conditions, ground vegetation is an important indicator of long-term soil conditions and their spatial variability. Thus it can be used as a rough *in situ* estimate of the CH$_4$ flux when planning the locations of the measurement plots, as demonstrated by this study. The Pleurozium-group was the most prevalent vegetation type within our sample points, representing typical ground vegetation of boreal pine forests. The mean CH$_4$ flux of the sample points in the Pleurozium-group was close to the modelled seasonal mean fluxes.

Even though the difference in mean soil moisture between the studied seasons was not large, the wet areas were wetter and wider in May–July than in August–October, resulting in CH$_4$ emissions in May–July, while at the dry areas the CH$_4$ uptake increased substantially between the seasons. This suggests that either the activity of CH$_4$ oxidizing bacteria seems to increase towards autumn at the dry areas, which could be linked to soil temperature being at the highest level in August, or it may be that the activity of methanogens in the deeper soil (or microsites) decreases due to drying. Previously reported results indicate that there is a local optimal soil moisture for CH$_4$ oxidation, and in boreal forest soils the oxidation of CH$_4$ decreases when the soil moisture increases (Billings et al., 2000), whereas low soil water content as such does not remarkably decrease CH$_4$ uptake of boreal forests' mineral soils (Saari et al., 1998). The oxidation of CH$_4$ has been discovered to be at the lowest level in late spring and early summer, and the

most effective in autumn, by both the high affinity and low affinity methanotrophs (Reay et al., 2005). Similarly to our results, upscaling of $CH_4$ fluxes across hillslope transects in a temperate deciduous forest by Warner et al. (2019) demonstrated $CH_4$ emissions from low-elevation areas in early summer, and uptake in late summer – moreover, the magnitude of the upscaled fluxes were similar to our upscaling results. Contrarily, Aaltonen et al. (2011) found the $CH_4$ uptake to be stronger in May–June compared to late summer and autumn at the hilltop of the SMEAR II site (in 2008) (Aaltonen et al., 2011).

Our results at the hilltop showed higher $CH_4$ uptake (mean –8.01 µmol m$^{-2}$ h$^{-1}$ in May–July and –10.3 µmol m$^{-2}$ h$^{-1}$ in August–October) compared to the previous forest floor $CH_4$ flux measurements from the same hilltop area, reporting mean $CH_4$ flux of −7.0 µmol m$^{-2}$ h$^{-1}$ (between August 2006 and June 2007) (Skiba et al., 2009), and −4.6 µmol m$^{-2}$ h$^{-1}$ (between April–November in 2008) (Aaltonen et al., 2011). This may be explained by stronger $CH_4$ uptake due to drier years. Flux data processing techniques may also cause discrepancies between this and prior studies, as the widely used linear flux calculation method has been demonstrated to underestimate the $CH_4$ fluxes by on average 33% in a chamber inter-comparison study (Pihlatie et al., 2013). Skiba et al. (2009) and Aaltonen et al. (2011) used linear flux calculation method, whereas here we mainly used a non-linear method. On a wider perspective, the mean $CH_4$ fluxes obtained from both the measurements (mean of all the sample points: May–July – 5.07, August–October −8.67 µmol m$^{-2}$ h$^{-1}$) and the modelling (May–July −7.74, August–October −10.0 µmol m$^{-2}$ h$^{-1}$) in our study were in line with previously reported forest floor $CH_4$ fluxes from boreal and temperate coniferous forests (−0.62 to −15 µmol $CH_4$ m$^{-2}$ h$^{-1}$, Jang et al. 2006).

**4.2 Hot spots of $CH_4$ emissions**

At the site, there was one anomalous water-filled sample point (SW–W-3), from which the $CH_4$ emissions were at the same level as emissions reported from a nearby fen site (Li et al., 2016; Rinne et al., 2007, 2018). Even though the location of SW–W-3 had the highest water table level, some of the other sample points had also water table at or close to the soil surface. However, we did not measure as high $CH_4$ emissions from any of the other sample points – excluding SW–W-3, the highest emission was one order of magnitude smaller than the highest emission from SW–W-3. Out of ten highest emissions measured, seven were from SW–W-3, and the rest from sample points E–SE-4–6 at another peaty area. These substantially higher emissions from SW–W-3 compared to other sample points with high soil water content and equally high *Sphagnum* coverage may be related to joint effect of these two factors. The *Sphagnum* mosses thrive in wet conditions, where also $CH_4$ is produced, and most of the *Sphagnum* mosses growing in Finland are demonstrated to support methanotrophic bacteria (Larmola et al., 2010), which therefore naturally reduces the potential $CH_4$ emissions from *Sphagnum*-covered wet areas. However, it may be that while the *Sphagnum*-associated methanotrophs may reduce the $CH_4$ emissions from many of the sample points, they may not be active during the highest water level at SW–W-3.

Our results indicate that the observed spatial hot spots of $CH_4$ emissions seem to be prone to temporal variation, depending on the soil water status, affecting also the size of these patches. The measurement years being drier than the long term average (annual precipitation 576 mm in 2013 and 572 mm in 2014; average 711 mm for 1981–2010) suggests that the emission patches can be larger on wetter years. The temporal variability in soil moisture is mainly driven by meteorological forcing, and is suggested to be greater at locations with intermediate soil moisture than at the extremely dry or wet locations, and in the topsoil compared to deeper soil layers (Rosenbaum et al., 2012). In our study, the mean soil moisture decreased between the early summer and autumn in the two wettest vegetation groups,

but stayed at the same level in the two driest groups, while the mean $CH_4$ flux showed increasing uptake in all the vegetation groups between the two seasons. This demonstrates that the temporal changes in soil moisture affect mainly the wet areas of the forest in our study, while the driest areas tend to remain dry, probably due to the well-drained and shallow topsoil on top of a bedrock.

Based on our results, most of the wet plots were located at the areas with sandy or gravelly till as subsoil, while the areas with bedrock close to the soil surface (max. 1 m soil) had lower soil moisture. However, the sample points E–SE-4–6 were located in a depression with ca. 0.6 m deep peat layer, which was on top of a bedrock area according to the subsoil map. Praeg et al. (2017) have also reported bedrock type affecting the $CH_4$ flux, probably through soil properties, rooting of plants, plant species and microbial composition – however, for better understanding more research is needed.

**4.3 Representativeness of sample point locations**

In our study, the great advantage is the large amount of sample points, resolving the small-scale spatial variability of the forest floor $CH_4$ flux. Some previous studies have upscaled $CH_4$ flux using similar type of approaches across complex terrains with measurements from slope transects (Kaiser et al., 2018; Warner et al., 2019). The site studied here represents a typical commercial pine forest of the boreal areas, and the results are thus scalable to large area of similar type of forests in boreal zone. The mean $CH_4$ flux obtained from all the sample points was rather close but slightly higher than the upscaled $CH_4$ flux obtained from the model, indicating that the sample points covered the spatial variation of both the soil moisture and $CH_4$ flux at the area. However, while the mean values can be insufficient to tell a full story (i.e. it misses the spatial and temporal variability), comparison of means is important when targeting accurate landscape-scale $CH_4$ budget. Based on our results, we state that 15–20 sample points are needed to reliably cover an area of comparable size of a typical boreal commercial forest, however, demanding carefully-designed placement in order to cover the spatial heterogeneity. Yet an area with higher topographical variation may require more sample points. Our conclusion of the recommended number of sample points is slightly lower compared to the ICOS measurement protocol, which recommend to have at least 25 points at a site when using manual chambers (Pavelka et al., 2018). In addition, our results suggest that, in August–October, only three randomly picked sample points would be as representative sampling of the whole area as 15–20 sample points in May–July, which was probably due to smaller spatial variability in the $CH_4$ flux in autumn compared to early summer.

Usually, if no upscaling is implemented, the mean flux of the measurements is reported, neglecting the effect of the placement of the measurement points. In this study, the $CH_4$ flux measurements resulted in smaller mean uptake than the spatially modelled estimate for the whole area, even though 60 sample points were used, which emphasizes the importance of upscaling. Soil or forest floor $CH_4$ fluxes are most often studied using ca. 4–10 soil chambers per study site (Billings et al., 2000; Lohila et al., 2016; Savi et al., 2016; Skiba et al., 2009; Sundqvist et al., 2015), although a couple of studies so far applied 20 or more chambers (Dinsmore et al., 2017; Matson et al., 2009; Wang et al., 2013b; Warner et al., 2018). Upscaling or mean flux is therefore often based on assumption that the soil conditions are rather homogenous over the area, and/or that the heterogeneity is well represented by a small number of chambers (Sundqvist et al., 2015). The locations of the sample points should be selected based on the spatial variability of the driving parameters. This could be done e.g. by evaluating different topographic or remote-sensing-derived indices in the study area during the planning phase of a scientific experiment, so that the measurements cover the full range of flux drivers based on a priori knowledge. Together with long time series, it is critically important to cover the spatial

variability within different ecosystems, and link the $CH_4$ fluxes to landscape parameters in order to achieve more accurate estimations of $CH_4$ (and other GHG) fluxes over large areas. The vastly developed and increasingly common elevation-mapping methods can be highly practical for upscaling the $CH_4$ fluxes of different areas. Furthermore, Ueyama et al. (2018) found that wet $CH_4$ emission patches were important at a larch plantation, and could have a strong contribution to the canopy-scale fluxes – in our case we cannot fully conclude the impact at the ecosystem-scale, as the above-canopy fluxes were not included.

### 4.4 Upscaling of the $CH_4$ flux

The predictive performance of the RF model developed for $CH_4$ flux upscaling was in the same range or lower than in some prior studies (Kaiser et al., 2018; Warner et al., 2019) using topographic data to upscale $CH_4$ fluxes. It must be noted, however, that direct comparison of cross-validation results between studies is hampered by the different cross-validation techniques used, since the method used for cross-validation has an influence on the results (Roberts et al., 2017). Nevertheless, the cross-validation results indicate that a significant proportion of the $CH_4$ flux variability was not explained by the RF model, suggesting that important predictors were missing from the RF model development. These likely include variables linked to plant activity and/or soil organic carbon storage, since these are related to the amount of substrates available for the microbes. Even though we created the model based on the correlations with several potential drivers (Sect. 2.7), all the examined variables were not available in fine enough resolution to be accurate for the sample points (e.g. soil type), or were not directly available for the whole area (e.g. soil temperature, vegetation type), and thus we cannot fully conclude that these drivers would not affect the spatial variability of the $CH_4$ flux and improve the model performance. Remote-sensing-derived indices (e.g. NDVI) might be helpful, but these are either not available at spatial scales used in this study, or separately for forest floor. Soil moisture and $CH_4$ flux were not measured at the same time at all sampling locations, and hence despite using temporally averaged data there may have been some apparent spatial variability in the temporally averaged data due to unsynchronised sampling (e.g. some locations measured more during rainy days and others during sunny days). This apparent variability cannot be explained with the static topographic properties, and hence could partly explain the significant proportion of the $CH_4$ flux and soil moisture variability unexplained with the RF models.

Variability in the emissions from wet mineral soils has been estimated to explain most of the total interannual variability in $CH_4$ emissions globally (Spahni et al., 2011). Kaiser et al. (2018) reported that when the soil moisture was above 0.43 $m^3$ $m^{-3}$ the soil was a source of $CH_4$, while soil moisture below 0.38 $m^3$ $m^{-3}$ resulted in $CH_4$ sink. In our study, the soil moisture limit for the $CH_4$ emissions was similar, at 0.39 $m^3$ $m^{-3}$, while simultaneously (in May–July) areas with soil moisture as high as 0.73 $m^3$ $m^{-3}$ indicated $CH_4$ uptake. Thus, in the upscaling method we used, the cells with high soil moisture had both uptake and emission $CH_4$ flux values, which ultimately results from the measured data (Appendix A, Fig. A8).

It should be noted, however, that the modelled fluxes represent only the average $CH_4$ flux spatial pattern during the two seasons, and hence they do not capture the short-term temporal variability in the $CH_4$ fluxes caused by rapid variability of soil moisture inflicted e.g. by rain. Hence, the soil moisture may be occasionally wetter than the modelled moisture at some locations of the study domain, and therefore larger areas can be emitting $CH_4$ during (short) wet periods. For instance, Rosenbaum et al. (2012) showed with spatially extensive and continuous soil moisture measurements that intense precipitation events were significantly altering the soil moisture spatial variability in their study. Based on the continuous soil moisture data measured at the SMEAR II, there was a peak in soil moisture

in mid-April (Fig. 2) during snowmelt, when we only measured the flux at the hilltop. Thus, presumably the $CH_4$ emissions were highest in the beginning of May, when the soil temperature started to increase and the soil moisture was still high, and thus the spring emissions may have been even higher than observed. In order to capture accurately the temporal variability, and to avoid underestimation of the highest values, soil moisture should be monitored continuously with high spatial (hilltops, depressions, slopes) and temporal frequency for future upscaling research. Furthermore, comprehensive annual measurements of $CH_4$ flux are also needed, as the non-growing season fluxes are noted to have an important contribution to annual $CH_4$ flux, especially at upland sites (Treat et al., 2018).

**5 Conclusions**

The $CH_4$ fluxes of the boreal forest floor are spatially highly heterogeneous, including potential emission hotspots. Soil moisture and vegetation type are important drivers of the spatial variability of the $CH_4$ flux, and the spatial variability of these drivers should be taken into account already in the experimental planning. Furthermore, to obtain spatially reliable estimates, the fluxes should be upscaled using appropriate geospatial tools. Spatially extensive measurements and high-resolution modelling can help to further improve our understanding on the $CH_4$ dynamics of forests. Moreover, resolving the $CH_4$ flux over large spatial scale with high temporal frequency would be of great importance in order to reveal the variation between years and seasons. Eventually this should lead to more precise global $CH_4$ budget.

**Author contributions**

M.P. had the original idea of the study. E.V. conducted the field measurements, analysed the flux data, and was the main author of the paper. E.-S.T. conducted the TWINSPAN analysis. O.P., A.-J.K., V.K. and E.V. planned the modelling, and O.P. conducted the modelling and wrote the modelling parts of the paper. All authors discussed and commented the paper, and contributed to the writing.

The authors declare that they have no conflict of interest.

**Data availability statement**

The upscaled $CH_4$ flux and soil moisture data, their uncertainties, the spatial drivers used for the model, as well as the sample point means of the measurements are available at Zenodo (http://doi.org/10.5281/zenodo.4382801, Vainio et al. 2020).

**Acknowledgements**

Funding was provided by the Emil Aaltonen Foundation, the Academy of Finland (project numbers 294088, 288494, 2884941, 296116, 307192, 311970, 315424, 287039), and the European Research Council (ERC) under the European Union's Horizon 2020 research and innovation programme under grant agreement No 757695.

For research facilities we thank Academy of Finland Centre of Excellence (programmes 1118615, 307331, 272041) and ICOS-Finland (grant 281255). We thank the technical staff of Hyytiälä SMEAR II station for technical assistance, as well as Kira Ryhti, Hanna El-khouri, Mari Mäki, and Lucas Toniolo Junior for field work assistance.

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

895

# APPENDIX A

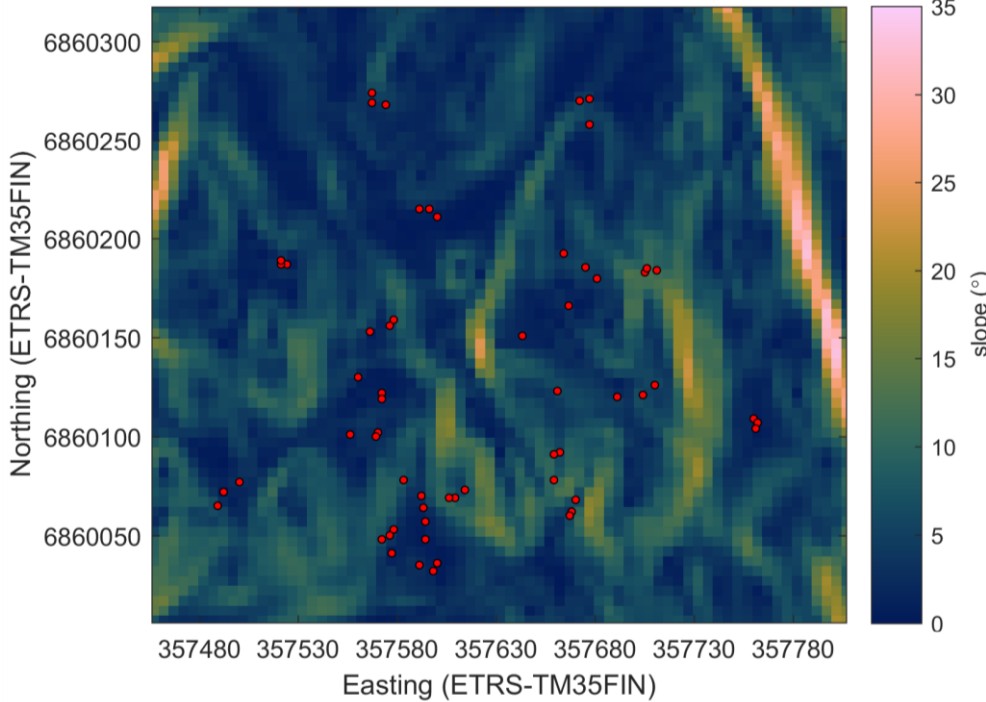

Figure A1. Slope at the study area. The red circles show the sample plots.

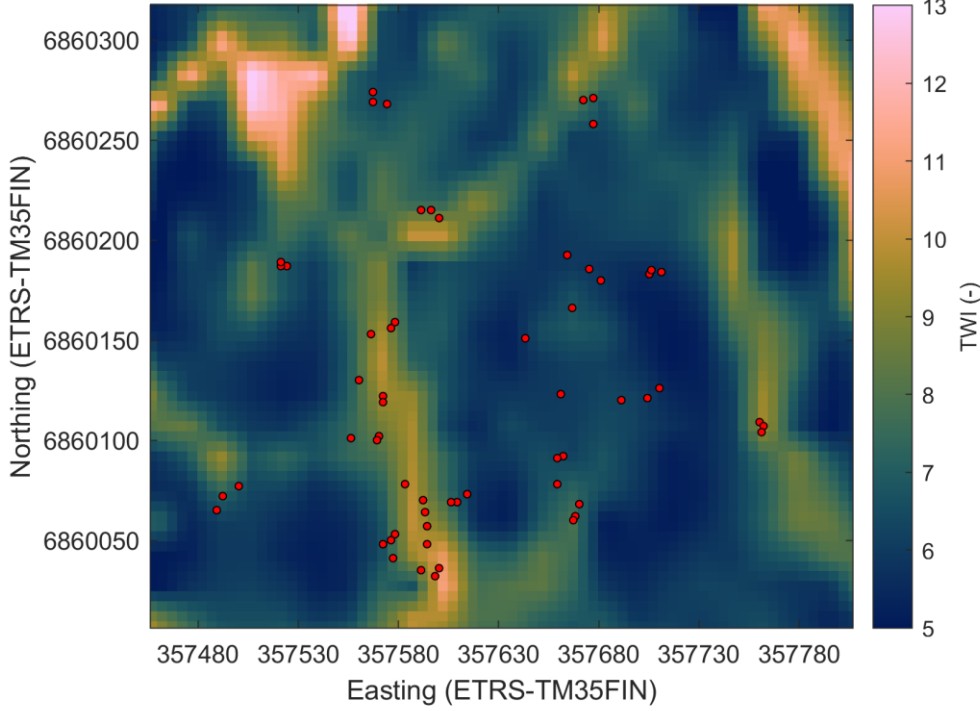

Figure A2. Topographic wetness index (TWI) at the study area. The red circles show the sample plots.

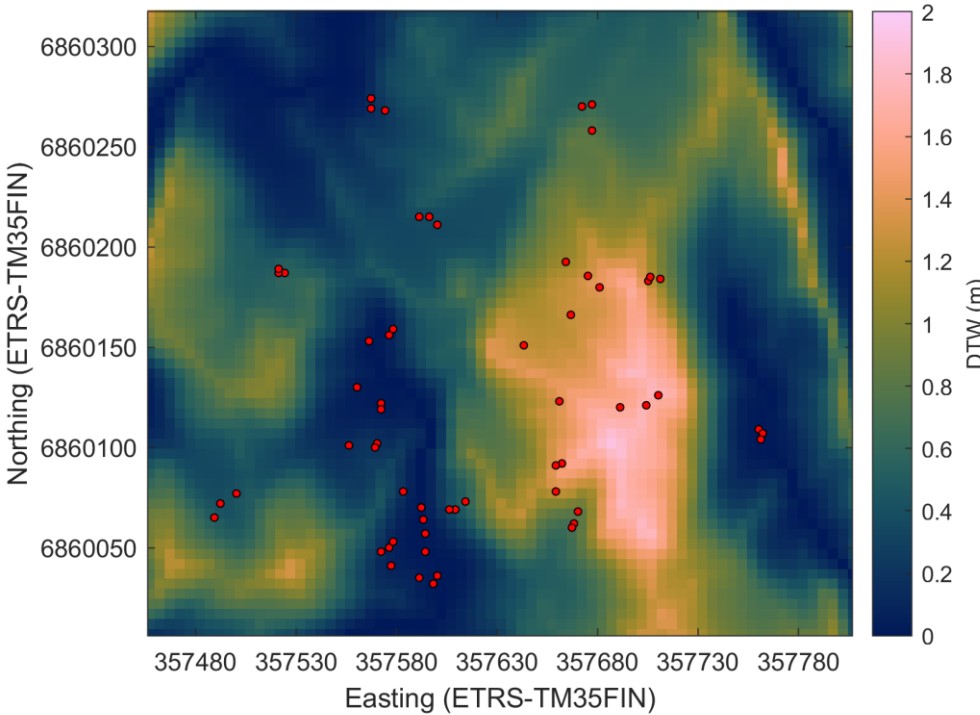

Figure A3. Cartographic depth-to-water index (DTW) at the study area. The red circles show the sample plots.

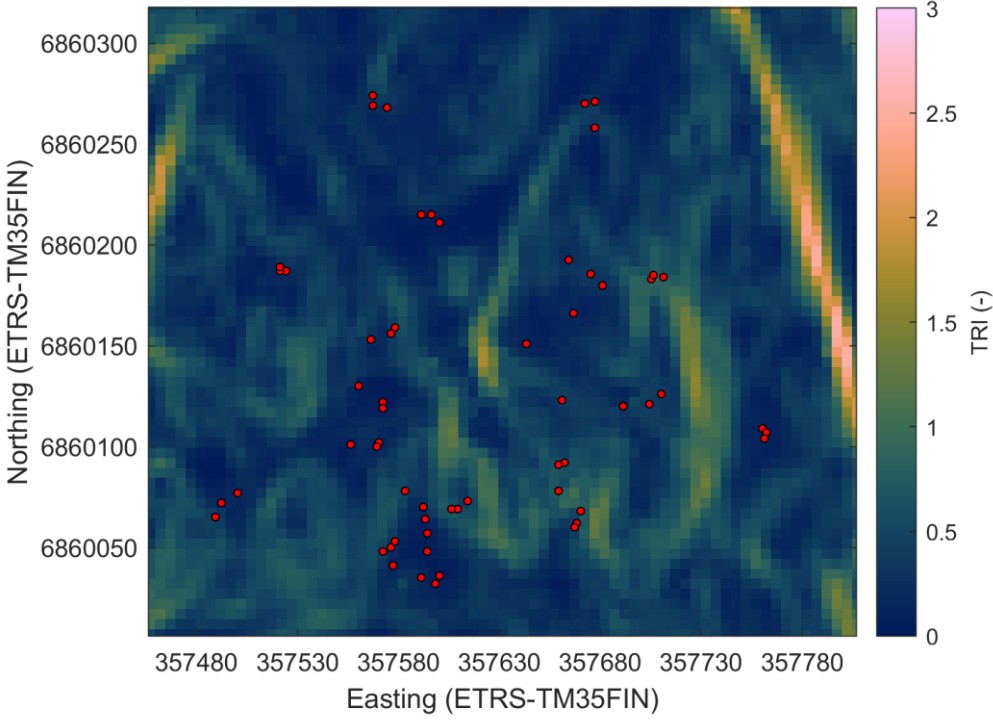

Figure A4. Terrain ruggedness index (TRI) at the study area. The red circles show the sample plots.

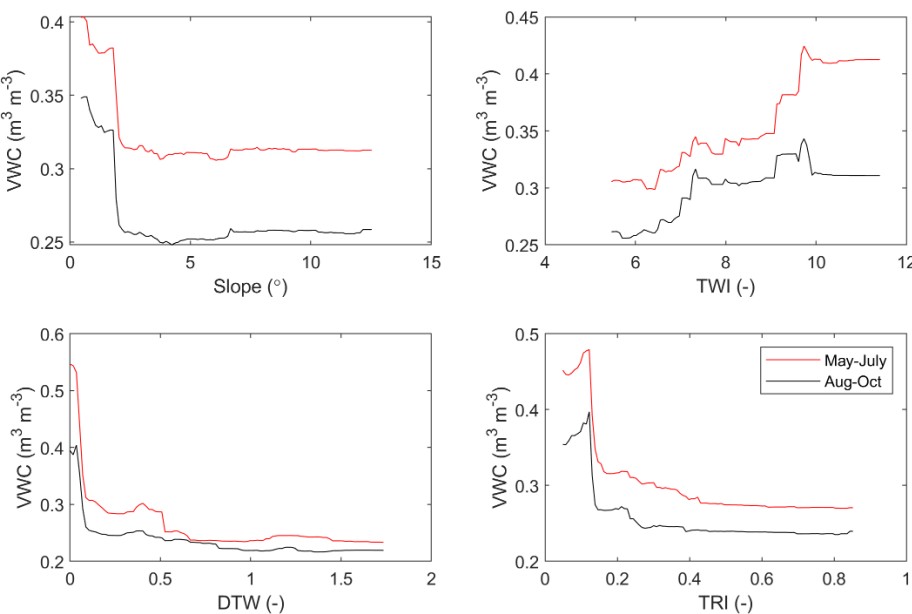

Figure A5. Partial dependences between the soil volumetric water content (VWC) and its drivers selected for the modelling (slope, topographic wetness index TWI, cartographic depth-to-water index DTW, and terrain ruggedness index TRI).

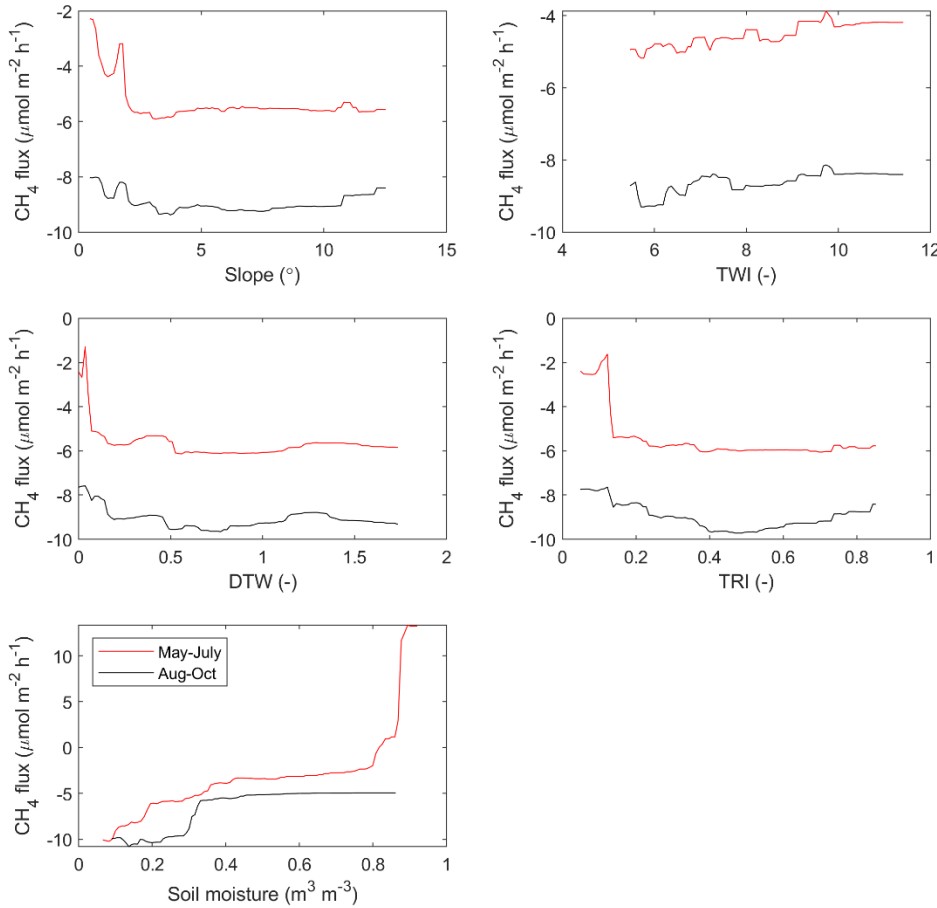

Figure A6. Partial dependences between the forest floor CH₄ flux and the driving parameters selected for the model (slope, topographic wetness index TWI, cartographic depth-to-water index DTW, terrain ruggedness index TRI, and soil volumetric water content).

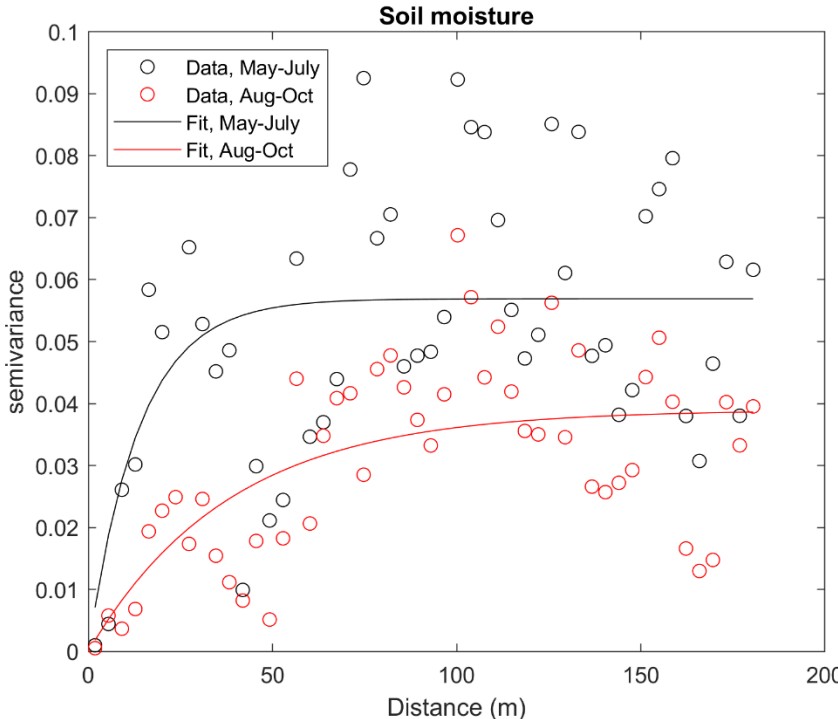

Figure A7. Empirical semivariogram (dots) and variogram model (line) for the soil moisture in May–July and August–October.

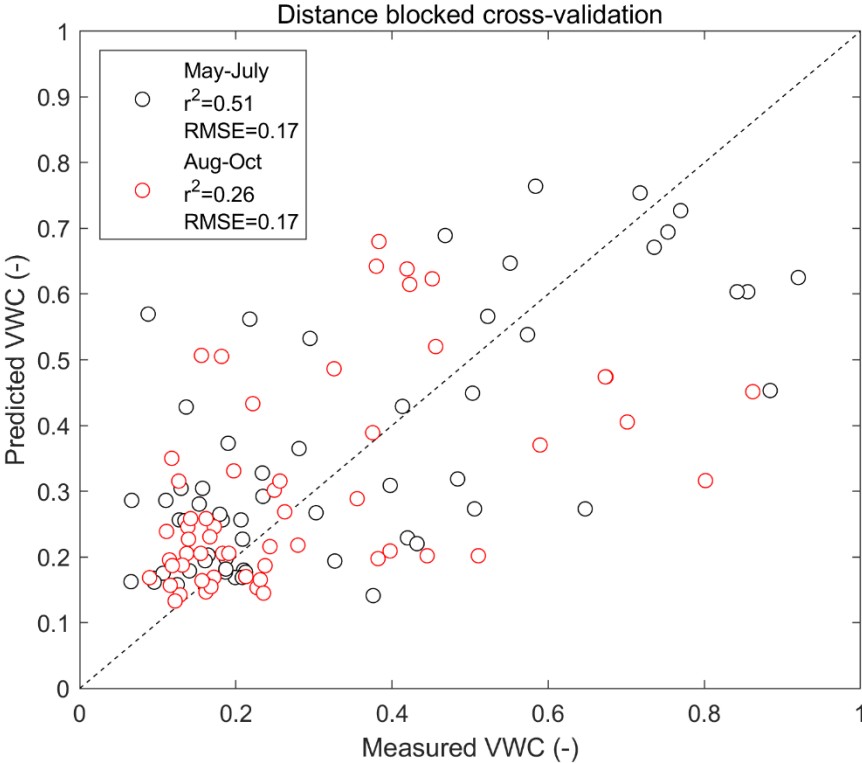

Figure A8. Distance-blocked cross-validation between the measured and predicted soil volumetric water content (VWC).

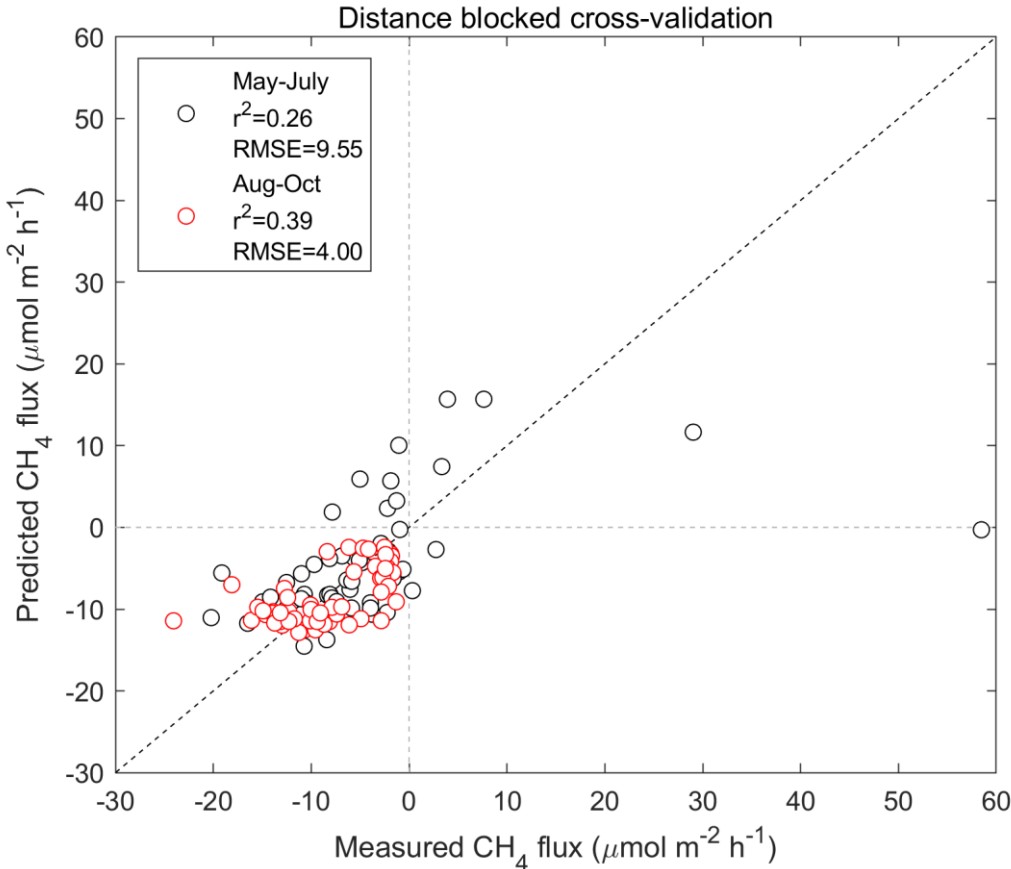

Figure A9. Distance-blocked cross-validation between the measured and predicted forest floor CH₄ flux.

Table A1. Spearman's correlation coefficients ($r_s$) and p-values for sample plot means. For the correlations between the $CH_4$ flux and soil temperature, the absolute values of flux were used.

| | | CH₄ flux | Soil Moisture | Soil Temp. | CH₄ flux May-July | CH₄ flux Aug-Oct | Soil Moist. May-July | Soil Moist. Aug-Oct | Soil Temp. May-July | Soil Temp. Aug-Oct | TWI | DTW | slope | TRI | Pine volume | Spruce volume | Birch volume | Spruce biomass | Pine biomass | Broadleafs biomass |
|---|---|---|---|---|---|---|---|---|---|---|---|---|---|---|---|---|---|---|---|---|
| CH₄ flux | $r_s$ | 1 | | | | | | | | | | | | | | | | | | |
| | p | | | | | | | | | | | | | | | | | | | |
| Soil Moisture | $r_s$ | 0.78 | 1 | | | | | | | | | | | | | | | | | |
| | p | <0.001 | | | | | | | | | | | | | | | | | | |
| Soil Temperature | $r_s$ | 0.06 | 0.03 | 1 | | | | | | | | | | | | | | | | |
| | p | >0.05 | >0.05 | | | | | | | | | | | | | | | | | |
| CH₄ flux May-July | $r_s$ | - | - | - | 1 | | | | | | | | | | | | | | | |
| | p | - | - | - | | | | | | | | | | | | | | | | |
| CH₄ flux Aug-Oct | $r_s$ | - | - | - | 0.83 | 1 | | | | | | | | | | | | | | |
| | p | - | - | - | <0.0001 | | | | | | | | | | | | | | | |
| Soil Moist. May-July | $r_s$ | - | - | - | 0.83 | - | 1 | | | | | | | | | | | | | |
| | p | - | - | - | <0.0001 | - | | | | | | | | | | | | | | |
| Soil Moist. Aug-Oct | $r_s$ | - | - | - | - | 0.63 | 0.87 | 1 | | | | | | | | | | | | |
| | p | - | - | - | - | <0.0001 | <0.0001 | | | | | | | | | | | | | |
| Soil Temp. May-July | $r_s$ | - | - | - | 0.25 | - | -0.133 | - | 1 | | | | | | | | | | | |
| | p | - | - | - | 0.06 | - | 0.310 | - | | | | | | | | | | | | |
| Soil Temp. Aug-Oct | $r_s$ | - | - | - | - | -0.31 | - | 0.14 | 0.21 | 1 | | | | | | | | | | |
| | p | - | - | - | - | 0.016 | - | 0.28 | 0.11 | | | | | | | | | | | |
| TWI | $r_s$ | 0.53 | 0.65 | 0.04 | 0.51 | 0.48 | 0.65 | 0.67 | -0.09 | 0.01 | 1 | | | | | | | | | |
| | p | <0.0001 | <0.0001 | >0.05 | <0.0001 | <0.001 | <0.0001 | <0.0001 | >0.05 | >0.05 | | | | | | | | | | |
| DTW | $r_s$ | -0.54 | -0.67 | -0.05 | -0.56 | -0.50 | -0.72 | -0.70 | 0.10 | 0.01 | -0.74 | 1 | | | | | | | | |
| | p | <0.0001 | <0.0001 | >0.05 | <0.0001 | <0.0001 | <0.0001 | <0.0001 | >0.05 | >0.05 | <0.0001 | | | | | | | | | |
| slope | $r_s$ | -0.51 | -0.61 | -0.03 | -0.50 | -0.49 | -0.61 | -0.55 | 0.07 | -0.13 | -0.58 | 0.59 | 1 | | | | | | | |
| | p | <0.0001 | <0.0001 | >0.05 | <0.0001 | <0.0001 | <0.0001 | <0.0001 | >0.05 | >0.05 | <0.0001 | <0.0001 | | | | | | | | |
| TRI | $r_s$ | -0.53 | -0.64 | -0.04 | -0.52 | -0.49 | -0.66 | -0.57 | 0.01 | -0.09 | -0.56 | 0.59 | 0.97 | 1 | | | | | | |
| | p | <0.0001 | <0.0001 | >0.05 | <0.0001 | <0.0001 | <0.0001 | <0.0001 | >0.05 | >0.05 | <0.0001 | <0.0001 | <0.0001 | | | | | | | |
| Pine volume | $r_s$ | 0.07 | -0.11 | 0.02 | -0.03 | 0.09 | -0.13 | -0.18 | 0.21 | 0.06 | -0.38 | 0.35 | 0.15 | 0.11 | 1 | | | | | |
| | p | >0.05 | >0.05 | >0.05 | >0.05 | >0.05 | >0.05 | >0.05 | >0.05 | >0.05 | <0.01 | <0.01 | >0.05 | >0.05 | | | | | | |
| Spruce volume | $r_s$ | -0.35 | -0.20 | 0.02 | -0.28 | -0.31 | -0.19 | -0.10 | -0.06 | 0.05 | -0.03 | -0.06 | 0.14 | 0.18 | -0.60 | 1 | | | | |
| | p | <0.01 | >0.05 | >0.05 | <0.05 | <0.05 | >0.05 | >0.05 | >0.05 | >0.05 | >0.05 | >0.05 | >0.05 | >0.05 | <0.0001 | | | | | |
| Birch volume | $r_s$ | -0.10 | -0.02 | -0.01 | 0.01 | -0.14 | 0.04 | 0.05 | -0.07 | -0.15 | 0.09 | -0.30 | 0.10 | 0.07 | -0.39 | 0.31 | 1 | | | |
| | p | >0.05 | >0.05 | >0.05 | >0.05 | >0.05 | >0.05 | >0.05 | >0.05 | >0.05 | >0.05 | <0.05 | >0.05 | >0.05 | <0.01 | <0.05 | | | | |
| Spruce biomass | $r_s$ | -0.35 | -0.13 | -0.01 | -0.28 | -0.31 | -0.11 | -0.07 | -0.04 | -0.05 | 0.05 | -0.16 | 0.07 | 0.09 | -0.63 | 0.95 | 0.38 | 1 | | |
| | p | <0.01 | >0.05 | >0.05 | <0.05 | <0.05 | >0.05 | >0.05 | >0.05 | >0.05 | >0.05 | >0.05 | >0.05 | >0.05 | <0.0001 | <0.0001 | <0.01 | | | |
| Pine biomass | $r_s$ | 0.11 | -0.11 | -0.03 | 0.01 | 0.14 | -0.12 | -0.20 | 0.08 | 0.09 | -0.36 | 0.32 | 0.10 | 0.07 | 0.95 | -0.66 | -0.39 | -0.69 | 1 | |
| | p | >0.05 | >0.05 | >0.05 | >0.05 | >0.05 | >0.05 | >0.05 | >0.05 | >0.05 | <0.01 | <0.05 | >0.05 | >0.05 | <0.0001 | <0.0001 | <0.01 | <0.0001 | | |
| Broadleafs biomass | $r_s$ | -0.16 | -0.07 | -0.09 | -0.09 | -0.17 | -0.04 | 0.01 | -0.07 | -0.28 | 0.15 | -0.31 | 0.10 | 0.11 | -0.38 | 0.18 | 0.84 | 0.31 | -0.37 | 1 |
| | p | >0.05 | >0.05 | >0.05 | >0.05 | >0.05 | >0.05 | >0.05 | >0.05 | <0.05 | >0.05 | <0.05 | >0.05 | >0.05 | <0.01 | >0.05 | <0.0001 | <0.05 | <0.01 | |