# Peer review of "Topography-based statistical modelling reveals high spatial variability and seasonal emission patches in forest floor methane flux"

_Biogeosciences, 2020_

## Referee Comment (RC1) · Anonymous Referee #1 · 18 Sep 2020

Vainio et al. studied the magnitude, spatiotemporal patterns, and drivers of forest floor CH4 flux. This study provides new insights into the mechanisms driving CH4 fluxes and utilizes a relatively novel methodological framework that focuses on spatial heterogeneity and statistical upscaling, therefore representing a substantial contribution to scientific progress. The manuscript is well written and has clear main aims. However, the Methods section still needs some technical clarifications and the structure of the Discussion could be revised to make sure it is easy to follow.

Here are a key few topics that I think still need to be clarified (more explanations in the line-specific comments):

[Figure]

1. The modeling framework

Line 225: How many observations did you have for May-July and August-October? Did you have many measurements from one point in your model (e.g., early May measurement, late July measurement)? Can you be certain that the soil moisture measurements conducted within one study period (e.g. in early May and late July) can be directly compared and used in the same model, even though the soils tend to get drier during the summer? I think it's important that the measurements that you use for your response variable (i.e., soil moisture) are fully comparable with each other.

225: Could you also describe why you decided to use soil moisture as a predictor of fluxes instead of using the different topographic indices directly? Also, did you consider creating a continuous vegetation type raster based on your vegetation classes and the gridded layers for the study domain? This could have been a useful predictor for CH4 fluxes as well.

256: Why didn't you use the similar framework that you used for soil moisture to predict CH4 fluxes with soil moisture? You could have created a RF (or a GLM/GAM or some other) model with the measured soil moisture as a predictor, and then used that model to predict fluxes across the landscape using the predicted soil moisture. And this could have been repeated over the different bootstrapped soil moisture maps to get CH4 flux uncertainty map as well.

240: Could you add the response graphs (partial dependence plots) describing the relationship of these indices and soil moisture to the Appendix?

2. Description of the model performance

Line 20: Somewhere here I would add a sentence about how the statistical models performed, and how reliable your results are.

377: I would be interested to see a scatterplot of the observed and predicted (CV) fluxes to see how well the model predicts high and low soil moisture values. Same

applies to CH4 flux (line 406).

542: Somewhere in the Discussion you should also discuss how well your upscaling performed compared to previous studies. What are the main uncertainties, and how can these uncertainties be reduced? What predictors are you missing? How about other RS-derived indices, such as NDVI?

Here are a few more minor suggestions to the manuscript:

Title: I would consider adding the word "statistical modelling" somewhere in the title

Line 19: I would say "using digital elevation model-derived topographic indices" instead of "topography"

28: I was a bit surprised to see this methodological suggestion as a final sentence concluding your study. I would consider changing this to something more broader, e.g. to the sentence on line 565-572.

71: "Large amount of measurement points" is a rather subjective statement as for some people this might mean hundreds or thousands of observations. Maybe define the rough amount of measurement points instead, and mention that this is more than has previously been used

75: With one driving parameter (i.e. soil moisture), right? You didn't have many driving parameters to make the upscaled CH4 flux map?

76: But what about Kaiser et al., 2018?

105: You could add an index map to this figure showing e.g. the location of Hyytiälä too

210: I would describe these gridded layers in their own paragraph, similar to the other environmental measurements, and dedicated this one to the models only.

220: Could you provided a little bit more information about what parameters were

chosen for the different indices? For example, TWI can change quite a bit depending on what parameters you use in the calculation.

255: Should be Aalto et al., (2018), not (Aalto et al., 2018)

370: This figure could be moved to the supplementary – it's not so useful for the reader because there are so many different points.

400: This is just an idea, but you could also replace these two maps by a map that describes the mean summer soil moisture and a map that describes its change over the growing season. It might be easier for the reader to spot the areas that are drying this way.

431: I would use the same color scheme that you used in Fig. 8 for the Fig. 9 as well, to make sure that you are using different color schemes for soil moisture and CH4 fluxes.

440: The discussion is rather long and without subtitles it is a little bit hard to follow. Could you consider adding a few subtitles and structuring it according to your main aims of the study (spatial variation, drivers and upscaling, hot spots)?

560: Again, I was a bit surprised to see this discussion here as it was not motivated in your introduction or it wasn't one of your main aims of the paper. Maybe include it to the introduction or remove it completely?

567: If you want to discuss the sampling strategy, I would provide some more details here. E.g., how should the sample points be selected (e.g. systematic grid, gradient, random sampling, researcher-defined)? What is the number of temporal replicates required to understand spatiotemporal variability in this system? Further, in the abstract you mention that capturing the environmental variability requires 15-20 sample points. But do you think using statistical methods (e.g. random forest) with 15-20 points is reliable?

---

## Referee Comment (RC2) · Anonymous Referee #2 · 23 Sep 2020

Recommendation: accept with major revisions

General Comments

This study investigated spatial variability of CH4 fluxes across a hilltop study site in a boreal forest, as well as their relationship to vegetation and soil moisture. The data generated by this study is very useful, as boreal forests are a critical ecosystem in global GHG dynamics. The methods for collecting the field measurements were rigorous and well done. However, 1) I have major issues with the upscaling approach and how it was used within the narrative of the manuscript. Additionally, 2) I am confused by the analysis of how many measurement points are necessary to model plot scale

fluxes.

1) What is the advantage of first modeling soil moisture and then using modeled soil moisture to model CH4 flux. Were TWI, slope, DTW, and TRI also evaluated for their relationships to CH4 flux, and could this be directly modeled without first modeling soil moisture? It seems that uncertainty is being compounded by introducing the uncertainty associated with the soil moisture-CH4 relationship as well as the RF model uncertainty. I understand that measuring soil moisture is logistically much easier than CH4, which would be useful for temporal and potentially spatial gap filling, but this study is only looking at average CH4 flux for two time periods. On top of this, there is never a discussion of why the approach of modeling soil moisture and then CH4 flux is advantageous. Furthermore, the authors dedicate a large portion of this manuscript to the upscaling exercise, but barely, if at all, discuss whole plot scale fluxes. It would be interesting to hear how much the estimated net CH4 sources offset the plot level sink between the two time periods, and how uncertain their plot level fluxes are. After all, the primary purpose of upscaling is not to accurately predict CH4 flux at every individual point, it is to enhance our predictive capability of large-scale CH4 exchange in a way that reflects soil heterogeneity.

2) The number of points analysis is highlighted in the abstract and results and discussion, but it is not mentioned in the methods. Unless I am mistaken, the conclusion is only based on the fact that there are similar means between the predictions and observations at N points. I do not agree with the authors on this and believe that substantially more work would be needed to demonstrate the necessary number of points. It would be more useful to randomly subsample the flux observations and build soil moisture-CH4 relationships from the random subsets. Then see how upscaled fluxes based on these relationships compare to the predictions made using the whole dataset. Finally, the manuscript struggles to clearly communicate model and equation uncertainty at many points, which I tried to note below. The authors should also report their modeled CH4 predicted fluxes for pixels corresponding to the sample sites, which would help

explain whether differences in upscaled fluxes are caused by a model bias or because of the heterogeneity of the predictor variable domain.

Specific Comments

Abstract: If possible, add some descriptive statistics (i.e. mean, min, max) for what CH4 fluxes were observed in each season.

20: No comma needed after flux.

21: The wording "as well as on the related ground vegetation" is confusing to me.

Introduction:

32: Remove word "also", perhaps provide estimated percentage contributions of each global sink for context.

33-35: I do not believe that this is the current paradigm. Observed methane fluxes are the net sum of both opposing processes occurring in the soil.

40: This is not an instant effect, however, which has implications on the influence of both total soil moisture and its temporal variability. It would be useful to note this here.

53-56: I think this is a great point, but I am confused why it is in this paragraph.

57-62: This paragraph is important but it could be written more clearly. Are the authors trying to say that we often consider ecosystem fluxes in large-scale models but have not adequately accounted for heterogenous sources/sinks within the ecosystem (which likely respond to environmental changes differently)?

Methods:

152: Is there an explanation for why one measurement would be so large?

177: It would be good to provide a +/- range for what types of temperature variation were observed here as justification.

220: Was the TWI then resampled/interpolated or left at coarse resolution? Additionally, I would hesitate to say that the TWI is "not accurate" at fine scales since it is simply a statistical metric and not a measurement of anything. It does, however, have limited application for estimating soil moisture on very high resolution DEMs because the metric itself is very sensitive to surface microtopography and noise.

225-242: I believe this paragraph could be rewritten to be better related to this study. A lot of the information on the inner workings of the RF algorithm can be condensed with an appropriate citation and a brief note on the advantages/disadvantages of RF over simpler techniques like multiple regressions. More information on why the model parameters (ntree = 300, mtry = 2) and predictor variables were selected would be helpful.

Results:

304-306: Having S.D. values or some indicator of variability in soil moisture besides these means would be helpful here and in other parts of the results section.

326-327: What was the data of the highest emission outlier? It could be nice to see where it and other CH4 measurements fall on the time series graphs above.

333-334 and Fig 3: This is unclear to me. Is this a temporally static correlation between the mean of all CH4 fluxes at each point and the mean of soil moisture at each point? 339: Is "September" supposed to be "October" here?

Fig 4: It would be nice to break these plots up by May-July and August-September observations.

361-367: Again, reporting the only the mean is limiting, also report SD (or some other metric of variability) within these sample groups.

Fig 5: Was this variability maintained between the early to late summer transition? It would be good to show both groups on this plot, but might make things too cluttered.

Table 2. It would be very helpful here to report the modeled statistics for both the whole area and at the sample points. Currently it is unclear whether the modeled soil moisture is systematically lower and therefore causing systematic overestimations of CH4 uptake, or if the domain of the entire study area happens to be drier on average leading to a larger estimated CH4 uptake.

Fig 7: Normalizing the uncertainty at each pixel by its predicted value would help communicate the spatial patterns in the consistency of the RF ensemble output. I would also suggest that the authors add a note on interpreting this uncertainty, which is more of a measurement of the agreement of predictions among multiple RF iterations than the error between predictions and observations like RMSE. I am a major supporter of reporting ensemble uncertainty along with model metrics like RMSE, but the wording can get very confusing!

418-421: This is another place where normalizing the uncertainty of the ensemble predictions is useful.

422-424: I may have missed it, but I do not remember seeing this approach described in the methods and it is kind of unclear here. I am also confused by what this is supposed to demonstrate.

Discussion:

444: This is unclear. The RF model was just used to estimate spatial distributions of soil moisture, which were then used to predict CH4 flux based on a linear model, correct?

451-454: I am not sure what these lines are doing in this paragraph. They seem disconnected from the point.

457: How are these two species different in terms of phenology, growth form, and root structure? If they are similar, I would hesitate to infer that the vegetation is affecting CH4 flux rather than soil properties other than moisture.

469-470: Yes, but variability within point clusters was not communicated to the readers. It would be very useful to include.

470-472: I do not agree with this. The points created the domain of the training data, so we would expect the model output to be constrained by that domain. Additionally, the mean of the data only tells part of the story. It would be much more useful to compare the distributions of prediction values vs. observations.

494-496: This is interesting, what differences in the ecosystems/soil types may account for this?

503-509: It would be useful to communicate whole plot scale CH4 flux estimates, but net sums and total source and sink strength.

513-514: This is could also be due to reduced activity of methanogens in deeper soil layers/microsites.

565-566: Good point. Not only is it that the heterogeneity is well-represented, the sample set must also account for the relative coverage of landscape features for a mean value to be accurate.

568-571: Yes. But why did this study focus on modeling soil moisture and not directly modeling CH4 flux based on landscape features?

Conclusions: This section could be filled out more completely. Differences in CH4 flux based on vegetation type was an interesting finding, for example.

---

## Author Comment (AC1) · 22 Oct 2020

We answer here shortly the main comments pointed out by the Referee #1. We will answer all the rest of the comments in detail later, as well as get back to the suggested modifications.

Referee #1 1. The modeling framework "Line 225: How many observations did you have for May-July and August-October? Did you have many measurements from one point in your model (e.g., early May measurement, late July measurement)? Can you be certain that the soil moisture measurements conducted within one study period (e.g. in early May and late July) can be directly compared and used in the same model,

even though the soils tend to get drier during the summer? I think it's important that the measurements that you use for your response variable (i.e., soil moisture) are fully comparable with each other."

AC: There were around 6 measurements (median) from each sample point in both of the seasons. Our data is not temporally very comprehensive, and thus we opted to do two static periods. Still, we wanted to take into account that the soil does, in fact, get drier towards autumn. We assume that this two-seasons-strategy is good compromise with the data we have. We actually started the measurements in late May (line 132), so we missed the wettest period in the spring, and the most active measurement period was in June–August (line 135). However, we still ended up having approximately same amount of data for both seasons.

"225: Could you also describe why you decided to use soil moisture as a predictor of fluxes instead of using the different topographic indices directly? Also, did you consider creating a continuous vegetation type raster based on your vegetation classes and the gridded layers for the study domain? This could have been a useful predictor for CH4 fluxes as well."

AC: We opted to use this two-step modelling scheme because based on initial testing soil moisture was the most important explanatory variable for CH4 fluxes, and the other explanatory variables (i.e. topography indices) had only a marginal impact. Further, we assumed that the correlation between the CH4 flux and the topography indices was only due to the fact that they both correlated with moisture. Hence, the two-step modelling approach was likely less prone to spurious correlations between drivers and response variable. However, due to the referee comment below, we will re-evaluate our approach and consider modelling CH4 fluxes directly with the RF technique and using soil moisture as a driver in the model. We considered but did not create a continuous vegetation type raster, because there are many drivers affecting the vegetation, and we did not have data of all such drivers. There is no direct connection between e.g. vegetation and soil moisture (Fig. 4), nor with topography. A continuous vegetation

type raster would have probably required more thorough mapping of ground vegetation of the entire area.

"256: Why didn't you use the similar framework that you used for soil moisture to predict CH4 fluxes with soil moisture? You could have created a RF (or a GLM/GAM or some other) model with the measured soil moisture as a predictor, and then used that model to predict fluxes across the landscape using the predicted soil moisture. And this could have been repeated over the different bootstrapped soil moisture maps to get CH4 flux uncertainty map as well."

AC: Thank you for this comment, this sounds like a good alternative for the approach now used in the manuscript. We will re-evaluate our modelling approach and consider changing it to follow this referee suggestion.

"240: Could you add the response graphs (partial dependence plots) describing the relationship of these indices and soil moisture to the Appendix?"

AC: Yes, we can add these.

---

## Author Comment (AC2) · 22 Oct 2020

We answer here shortly the main comments pointed out by the Referee #2. We will answer all the rest of the comments in detail later, as well as get back to the suggested modifications.

Referee #2: There are two larger general comments: 1) Concerning the upscaling method:

AC: Initially, we tested how the different spatial explanatory variables are affecting on how well the model can predict the CH4 fluxes. Soil moisture was the most important

explanatory variable, while the other variables had only a marginal impact on the model performance. Thus, we opted the selected approach of modelling soil moisture first, and using that for the CH4 flux. Other variables, such as slope or the wetness-indices, could not predict the CH4 flux as well directly. However, due to the Referee #1 comment, we will re-evaluate our modelling approach and consider modelling CH4 fluxes directly with RF technique with soil moisture included as a driver.

"On top of this, there is never a discussion of why the approach of modeling soil moisture and then CH4 flux is advantageous. Furthermore, the authors dedicate a large portion of this manuscript to the upscaling exercise, but barely, if at all, discuss whole plot scale fluxes. It would be interesting to hear how much the estimated net CH4 sources offset the plot level sink between the two time periods, and how uncertain their plot level fluxes are. After all, the primary purpose of upscaling is not to accurately predict CH4 flux at every individual point, it is to enhance our predictive capability of large-scale CH4 exchange in a way that reflects soil heterogeneity."

AC: Please see our main responses to Referee #1 for the reasoning for using this two-step modelling approach. We agree with the referee that the primary purpose of upscaling is to get an accurate estimate of landscape fluxes and not at individual points. We will revise the text so that this message gets across to the reader clearly, and add discussion on the whole area flux and the modelling, decreasing the proportion of some smaller details in the discussion. This was actually one of the main reasons why the analysis on chamber location bias was done (Fig. 10). Typically, mean of CH4 fluxes observed at a handful of chamber locations is reported and considered as representative of ecosystem CH4 exchange, however, this neglects any bias stemming from non-representative sampling locations. By using mean upscaled CH4 flux as a reference, we were able to show that 15-20 randomly selected chamber measurement locations (out of 60 locations available) were able to produce as accurate estimate of the landscape CH4 flux as averaging over all the chamber data. This information should prove useful when designing future chamber measurement campaigns in similar

locations aiming to achieve accurate landscape-level flux estimates without upscaling with e.g. machine learning techniques.

2) "It would be more useful to randomly subsample the flux observations and build soil moisture-CH4 relationships from the random subsets. Then see how upscaled fluxes based on these relationships compare to the predictions made using the whole dataset."

AC: We acknowledge that the wording on lines 422-425 in the manuscript needs clarification, and this has maybe caused misunderstanding. The idea of the analysis shown in Fig. 10 in the manuscript was to evaluate how many chamber measurement locations were needed to get an accurate estimate of landscape-level flux by only averaging over the measured chamber data without any upscaling with RF. Here the mean upscaled CH4 flux was used as a reference since it accounts for the soil heterogeneity (see above). It was shown that average over 15-20 locations resulted in a similar bias as average over all the chamber measurement locations. This should be useful information for future chamber measurements in similar locations. We will revise the text so that this analysis is clearer and describe the methodology briefly also in Materials and Methods (Sect. 2).

"The authors should also report their modeled CH4 predicted fluxes for pixels corresponding to the sample sites, which would help explain whether differences in upscaled fluxes are caused by a model bias or because of the heterogeneity of the predictor variable domain."

AC: Thank you for this comment, we will evaluate this.

Specific comments:

AC: Overall, here are good suggestions on how to improve the paper. We will try out most of the suggested improvements, as well as add the SD values in text with the reported mean values. The CH4 highest emission (1080 $\mu$mol m$-2$ h$-1$) was detected

on 5 June 2013 from chamber SW–W-3. This measurement was reported in the results, but omitted from further data analysis, even though there was no indication of any error in the measurement. The sample point was located on small water pond, where the water table level was most of the time above the peat surface. The CH4 emissions from this sample point were at the same level with typical peatland emissions. One possible explanation for the highest emission would be ebullition. Fig. 3 is indeed the correlation between the mean of all CH4 fluxes at each point and the mean of soil moisture at each point.

---

## Author Comment (AC3) · 9 Nov 2020

Here we answer all the comments by the Referee #1 in detail.

1. The modeling framework

RC: Line 225: How many observations did you have for May-July and August-October? Did you have many measurements from one point in your model (e.g., early May measurement, late July measurement)? Can you be certain that the soil moisture measurements conducted within one study period (e.g. in early May and late July) can be directly compared and used in the same model, even though the soils tend to get drier

during the summer? I think it's important that the measurements that you use for your response variable (i.e., soil moisture) are fully comparable with each other.

AC: There were around 6 measurements (median) from each sample point in both of the seasons. Our data is not temporally very comprehensive, and thus we opted to do two static periods. Still, we wanted to take into account that the soil does, in fact, get drier towards autumn. We assume that this two-seasons-strategy is good compromise with the data we have. We actually started the measurements in late May (line 132), so we missed the wettest period in the spring, and the most active measurement period was in June–August (line 135). However, we still ended up having approximately same amount of data for both seasons.

RC: 225: Could you also describe why you decided to use soil moisture as a predictor of fluxes instead of using the different topographic indices directly? Also, did you consider creating a continuous vegetation type raster based on your vegetation classes and the gridded layers for the study domain? This could have been a useful predictor for CH4 fluxes as well.

AC: We re-evaluated our modelling approach and will change the modelling: we model the CH4 fluxes directly with the RF technique and using soil moisture as a driver in the model (see Fig.1 in this document). This change was motivated by the fact that bulk of the criticism from the reviewers was directed towards the CH4 flux modelling approach. The manuscript will be modified accordingly. We considered but did not create a continuous vegetation type raster, because there are many drivers affecting the vegetation, and we did not have data of all such drivers. There is no direct connection between e.g. vegetation and soil moisture (Fig. 4), nor with topography. A continuous vegetation type raster would have probably required more thorough mapping of ground vegetation of the entire area.

RC: 256: Why didn't you use the similar framework that you used for soil moisture to predict CH4 fluxes with soil moisture? You could have created a RF (or a GLM/GAM or

some other) model with the measured soil moisture as a predictor, and then used that model to predict fluxes across the landscape using the predicted soil moisture. And this could have been repeated over the different bootstrapped soil moisture maps to get CH4 flux uncertainty map as well.

AC: See the previous response, we changed our modelling approach to follow this referee suggestion.

RC: 240: Could you add the response graphs (partial dependence plots) describing the relationship of these indices and soil moisture to the Appendix?

AC: Yes, we can add these (see Figs. 2 and 3 in this document).

2. Description of the model performance

RC: Line 20: Somewhere here I would add a sentence about how the statistical models performed, and how reliable your results are.

AC: Yes, we can add this to the abstract.

RC: 377: I would be interested to see a scatterplot of the observed and predicted (CV) fluxes to see how well the model predicts high and low soil moisture values. Same applies to CH4 flux (line 406).

AC: We can add the scatterplot showing cross-validation results to the appendix (see Fig. 4 in this document).

RC: 542: Somewhere in the Discussion you should also discuss how well your up-scaling performed compared to previous studies. What are the main uncertainties, and how can these uncertainties be reduced? What predictors are you missing? How about other RS-derived indices, such as NDVI?

AC: According to our current impression, there are not many published studies trying to upscale CH4 fluxes from forest stand like ours. The previous study of (Sundqvist et al. 2015) used simple soil wetness and temperature relationship to upscale CH4

fluxes, whereas Kaiser et al., (2018) and Warner et al., (2019) studied more southern ecosystems. Furthermore, comparisons of e.g. cross-validation results are hampered by different cross-validation techniques used in different studies. In this manuscript, we utilized distance-blocked cross-validation since it is argued to produce more realistic estimates of cross-validation metrics than other techniques (Roberts et al., 2017). For instance, with traditional leave-one-out cross-validation the predictive performance of our RF model would seemingly improve (e.g. r2 increase from 0.51 and 0.26 to 0.67 and 0.56 for May–July and August–October, respectively). Therefore, we argue that direct comparison of cross-validation metrics between studies using different cross-validation strategies is not feasible. The discussion section of the manuscript does already discuss variables controlling CH4 fluxes, but we will modify the text so that it includes a section on how these predictors are missing in upscaling exercises. The scale where our measurements are carried out cannot be directly applied to similar spatial scale than remote sensing methods that is used to estimate NDVI, despite the rapid development of satellite products. However, we try include this to discussion.

Here are a few more minor suggestions to the manuscript:

RC: Title: I would consider adding the word "statistical modelling" somewhere in the title

AC: We consider to change the title to "Topography-based statistical modelling reveals . . ."

RC: Line 19: I would say "using digital elevation model-derived topographic indices" instead of "topography"

AC: We will modify this as suggested.

RC: 28: I was a bit surprised to see this methodological suggestion as a final sentence concluding your study. I would consider changing this to something more broader, e.g. to the sentence on line 565-572.

AC: We will modify this as suggested.

RC: 71: "Large amount of measurement points" is a rather subjective statement as for some people this might mean hundreds or thousands of observations. Maybe define the rough amount of measurement points instead, and mention that this is more than has previously been used.

AC: We will clarify this sentence in the following way: "In this study, we used relatively high number of measurement points (60 points on an area of ca. 10 ha) in order to fully cover the small-scale spatial variability in the $CH_4$ flux and its driving forces. Similar type of studies using chamber measurements are rarely based on more than 20 measurement points."

RC: 75: With one driving parameter (i.e. soil moisture), right? You didn't have many driving parameters to make the upscaled $CH_4$ flux map?

AC: Yes, originally we used soil moisture only. However, due to the Referee #1 comment we re-evaluated this approach and based on this re-evaluation opted to follow reviewer suggestion to use RF model also for $CH_4$ fluxes.

RC: 76: But what about Kaiser et al., 2018?

AC: Yes, probably good to be more precise, we will modify the text to: " Only a few studies (Kaiser et al., 2018; Warner et al., 2019) have applied similar approach, of which Kaiser et al. (2018) at a boreal coniferous forest, emphasizing the novelty of this study."

RC: 105: You could add an index map to this figure showing e.g. the location of Hyytiälä, too.

AC: Yes, this is a good suggestion.

RC: 210: I would describe these gridded layers in their own paragraph, similar to the other environmental measurements, and dedicated this one to the models only.

[Figure]

AC: Good suggestion, thank you, we will separate these under a new subheading.

RC: 220: Could you provided a little bit more information about what parameters were chosen for the different indices? For example, TWI can change quite a bit depending on what parameters you use in the calculation.

AC: TWI was calculated as a natural logarithm of the ratio between local upslope area draining through the point in question and tangent of the local slope. The upslope area was calculated using multiple flow direction algorithm of Freeman (1991), and local slope was calculated using adjacent points in DEM. The calculations were made with TopoToolbox, please see more details in Schwanghart and Kuhn (2010). We will add some more information on these to the methods.

RC: 370: This figure could be moved to the supplementary – it's not so useful for the reader because there are so many different points.

AC: We think that this figure gives a nice overview of the measured CH4 flux and its variation at the measurement points, and it is rather easy to see the spatial variation from this figure. (E.g. Fig. 4 is not giving this information, and a table would be more difficult to read.)

RC: 400: This is just an idea, but you could also replace these two maps by a map that describes the mean summer soil moisture and a map that describes its change over the growing season. It might be easier for the reader to spot the areas that are drying this way.

AC: This is an interesting thought and good to consider. However, we think that it might be more difficult for readers to understand what was done in this paper based on such a figure.

RC: 431: I would use the same color scheme that you used in Fig. 8 for the Fig. 9 as well, to make sure that you are using different color schemes for soil moisture and CH4 fluxes.

AC: Diverging colormaps (as the one used in Fig. 8) are suitable for data sets containing negative and positive values, because this way it is possible to emphasize the difference from zero. Hence, we opted to use a different colormap for Fig. 9 than Fig. 8 since random uncertainty cannot be negative. However, based on Referee #2 comment, we will modify Figs. 7 & 9 to show relative uncertainties, and as a result Fig. 9 colormap will be changed.

RC: 440: The discussion is rather long and without subtitles it is a little bit hard to follow. Could you consider adding a few subtitles and structuring it according to your main aims of the study (spatial variation, drivers and upscaling, hot spots)?

AC: Yes, this is a good suggestion, we will do this.

RC: 560: Again, I was a bit surprised to see this discussion here as it was not motivated in your introduction or it wasn't one of your main aims of the paper. Maybe include it to the introduction or remove it completely?

AC: This is a good remark. We will add shortly to the introduction that previously usually fewer measurement points have been used in soil chamber CH4 measurements, with the assumption that they are representative for a larger area.

RC: 567: If you want to discuss the sampling strategy, I would provide some more details here. E.g., how should the sample points be selected (e.g. systematic grid, gradient, random sampling, researcher-defined)? What is the number of temporal replicates required to understand spatiotemporal variability in this system? Further, in the abstract you mention that capturing the environmental variability requires 15-20 sample points. But do you think using statistical methods (e.g. random forest) with 15-20 points is reliable?

AC: Thank you for this comment. We will add shortly in this paragraph about the selection of sample points: in our opinion, e.g. the elevation maps would be useful when selecting the sample points. With this study we were not able to reveal more

high-frequency temporal variability, but we can add a sentence here that more frequent measurements would be needed for that – ideally it would require automatic chambers measuring e.g. once per day, or at minimum manual measurements every week. The idea of the analysis shown in Fig. 10 in the manuscript was to evaluate how many chamber measurement locations were needed to get an accurate estimate of landscape-level flux by only averaging over the measured chamber data without any upscaling with RF. This will be clarified in the manuscript (e.g. lines 422–425), as this was also commented by Referee #2.

References:

Freeman, T. G.: Calculating catchment area with divergent flow based on a regular grid, Computers & Geosciences, 17(3), 413–422. doi: 10.1016/0098-3004(91)90048-I, 1991.

Kaiser, K. E., McGlynn, B. L. and Dore, J. E.: Landscape analysis of soil methane flux across complex terrain, Biogeosciences, 15(10), 3143–3167, doi:10.5194/bg-15-3143-2018, 2018.

Roberts, D. R., Bahn, V., Ciuti, S., Boyce, M. S., Elith, J., Guillera-Arroita, G., Hauenstein, S., Lahoz-Monfort, J. J., Schröder, B., Thuiller, W., Warton, D. I., Wintle, B. A., Hartig, F. and Dormann, C. F.: Cross-validation strategies for data with temporal, spatial, hierarchical, or phylogenetic structure, Ecography (Cop.)., 40(8), 913–929, doi:10.1111/ecog.02881, 2017.

Schwanghart, W., and Kuhn N. J.: TopoToolbox: A set of Matlab functions for topographic analysis, Environ. Model. Softw., 25(6), 770–781, doi: 10.1016/j.envsoft.2009.12.002, 2010.

Sundqvist, E., Persson, A., Kljun, N., Vestin, P., Chasmer, L., Hopkinson, C. and Lindroth, A.: Upscaling of methane exchange in a boreal forest using soil chamber measurements and high-resolution LiDAR elevation data, Agric. For. Meteorol., 214–215,

393–401, doi:10.1016/j.agrformet.2015.09.003, 2015.

Warner, D. L., Guevara, M., Inamdar, S. and Vargas, R.: Upscaling soil-atmosphere $CO_2$ and $CH_4$ fluxes across a topographically complex forested landscape, Agric. For. Meteorol., 264, 80–91, doi:10.1016/j.agrformet.2018.09.020, 2019.

———————————————

[Figure]

[Figure]

**Fig. 1.** New Fig. 8: Maps of CH4 Flux with modified modelling approach.

[Figure]

**Fig. 2.** Partial dependence Soil moisture

[Figure]

**Fig. 3.** Partial dependence CH4 Flux

**Distance blocked cross-validation**

May-July
○ $r^2$=0.51
RMSE=0.17
Aug-Oct
○ $r^2$=0.26
RMSE=0.17

Predicted VWC (-)

Measured VWC (-)

**Fig. 4.** Distance blocked cross-validation

---

## Author Comment (AC4) · 9 Nov 2020

Here we answer all the comments by the Referee #2 in detail.

RC: There are two larger general comments: 1) Concerning the upscaling method:

AC: We re-evaluated our modelling approach based on these Referee comments, and will change the modelling to follow the suggestion by Referee #1: we model the CH4 fluxes directly with the RF technique and using soil moisture as a driver in the model (see Fig.1 in this document).

RC: On top of this, there is never a discussion of why the approach of modeling soil

moisture and then CH4 flux is advantageous. Furthermore, the authors dedicate a large portion of this manuscript to the upscaling exercise, but barely, if at all, discuss whole plot scale fluxes. It would be interesting to hear how much the estimated net CH4 sources offset the plot level sink between the two time periods, and how uncertain their plot level fluxes are. After all, the primary purpose of upscaling is not to accurately predict CH4 flux at every individual point, it is to enhance our predictive capability of large-scale CH4 exchange in a way that reflects soil heterogeneity.

AC: See the previous response, we changed the modelling approach.

We agree with the referee that the primary purpose of upscaling is to get an accurate estimate of landscape fluxes and not at individual points. We will revise the text so that this message gets across to the reader clearly, and add discussion on the whole area flux and the modelling, decreasing the proportion of some smaller details in the discussion. This was actually one of the main reasons why the analysis on chamber location bias was done (Fig. 10). Typically, mean of CH4 fluxes observed at a handful of chamber locations is reported and considered as representative of ecosystem CH4 exchange, however, this neglects any bias stemming from non-representative sampling locations. By using mean upscaled CH4 flux as a reference, we were able to show that 15-20 randomly selected chamber measurement locations (out of 60 locations available) were able to produce as accurate estimate of the landscape CH4 flux as averaging over all the chamber data. This information should prove useful when designing future chamber measurement campaigns in similar locations aiming to achieve accurate landscape-level flux estimates without upscaling with e.g. machine learning techniques.

RC: It would be more useful to randomly subsample the flux observations and build soil moisture-CH4 relationships from the random subsets. Then see how upscaled fluxes based on these relationships compare to the predictions made using the whole dataset.

BGD
AC: We acknowledge that the wording on lines 422-425 in the manuscript needs clarification, and this has maybe caused misunderstanding. The idea of the analysis shown in Fig. 10 in the manuscript was to evaluate how many chamber measurement locations were needed to get an accurate estimate of landscape-level flux by only averaging over the measured chamber data without any upscaling with RF. Here the mean upscaled CH4 flux was used as a reference since it accounts for the soil heterogeneity (see above). It was shown that average over 15-20 locations resulted in a similar bias as average over all the chamber measurement locations. This should be useful information for future chamber measurements in similar locations. We will revise the text so that this analysis is clearer and describe the methodology briefly also in Materials and Methods (Sect. 2).

RC: The authors should also report their modeled CH4 predicted fluxes for pixels corresponding to the sample sites, which would help explain whether differences in upscaled fluxes are caused by a model bias or because of the heterogeneity of the predictor variable domain.

AC: Thank you for this comment, we will evaluate this.

**Specific Comments**

RC: Abstract: If possible, add some descriptive statistics (i.e. mean, min, max) for what CH4 fluxes were observed in each season.

AC: Will be added.

RC: 21: The wording "as well as on the related ground vegetation" is confusing to me.

AC: Spelling mistake, will be corrected "as well as from the related ground vegetation".

RC: 33-35: I do not believe that this is the current paradigm. Observed methane fluxes are the net sum of both opposing processes occurring in the soil.

AC: While it is the prevailing paradigm that the availability of oxygen mainly controls

**BGD**
these processes in nature, and thus in general upland soils are no favourable place for CH4 production, there are possibilities for CH4 production taking place in these soils too, as we state in the next paragraph. The recent study of Treat et al. (2018) reported that during non-growing season upland soils can be a significant source of methane although they are generally considered as sinks during the growing season.

RC: 40: This is not an instant effect, however, which has implications on the influence of both total soil moisture and its temporal variability. It would be useful to note this here.

AC: We agree that the soils do not turn immediately sources of CH4 after inundation, but there are time lags between these processes. We will add this to the manuscript: "However, there are likely notable time lags between the start of inundation and methanogenesis, complicating the analyses of dependencies between these processes."

RC: 53-56: I think this is a great point, but I am confused why it is in this paragraph.'

AC: Thank you for this constructive comment but our impression is that this is logically introduced as it is.

RC: 57-62: This paragraph is important but it could be written more clearly. Are the authors trying to say that we often consider ecosystem fluxes in large-scale models but have not adequately accounted for heterogeneous sources/sinks within the ecosystem (which likely respond to environmental changes differently)?

AC: Yes, we are trying to say that the sources and sinks within the ecosystems are not adequately known or accounted. We will add a sentence in this paragraph, clarifying this.

Methods:

RC: 152: Is there an explanation for why one measurement would be so large?
AC: The sample point was located on small water pond, where the water table level was most of the time above the peat surface. The CH4 emissions from this sample point were at the same level with typical peatland emissions. One possible explanation for the highest emission would be ebullition.

RC: 177: It would be good to provide a +/- range for what types of temperature variation were observed here as justification.

AC: We will add these.

RC: 220: Was the TWI then resampled/interpolated or left at coarse resolution? Additionally, I would hesitate to say that the TWI is "not accurate" at fine scales since it is simply a statistical metric and not a measurement of anything. It does, however, have limited application for estimating soil moisture on very high resolution DEMs because the metric itself is very sensitive to surface microtopography and noise.

AC: Yes, TWI was then interpolated with bilinear interpolation to the finer grid (5x5). We agree, "not accurate" is not correct wording in this context. We will modify the text accordingly.

RC: 225-242: I believe this paragraph could be rewritten to be better related to this study. A lot of the information on the inner workings of the RF algorithm can be condensed with an appropriate citation and a brief note on the advantages/disadvantages of RF over simpler techniques like multiple regressions. More information on why the model parameters (ntree = 300, mtry = 2) and predictor variables were selected would be helpful.

AC: Based on this comment, we will condense the details related to the Random Forest algorithm. However, we believe that some description is needed since the method is not necessarily very familiar to the chamber flux measurement community. Minimum number of observations per tree leaf was set to 2, due to limited amount of data. Small number for this parameter helps when trying to capture dependencies at both BGD
ends of the training data distribution, especially when using limited dataset. Amount of trees in the RF model (ntree) was initially set to 300 based on prior experience with RF and value for this hyperparameter did not significantly influence results. However, the number of variables randomly sampled as candidates at each split (mtry) was not changed from its default value (one third of the total number of variables). The predictor variables were selected based on Spearman's rank correlation coefficient.

Results:

RC: 304-306: Having S.D. values or some indicator of variability in soil moisture besides these means would be helpful here and in other parts of the results section.

AC: We will add the SD values in text with the reported mean values.

RC: 326-327: What was the data of the highest emission outlier? It could be nice to see where it and other CH4 measurements fall on the time series graphs above.

AC: The CH4 highest emission (1080  $\mu$ mol m-2 h-1) was measured on 5 June 2013. We will add the date of the measurement in the text (line 326). It is unfortunately difficult to present an approachable timeseries plot of the measured CH4 fluxes due to high variation (large emissions vs. other data) – furthermore, we think there is no need to add any more figures to this paper.

RC: 333-334 and Fig 3: This is unclear to me. Is this a temporally static correlation between the mean of all CH4 fluxes at each point and the mean of soil moisture at each point?

AC: Yes.

RC: 339: Is "September" supposed to be "October" here?

AC: Thank you for pointing this out. These were actually calculated for May–Sep, which has been related to some previous version of the manuscript, but it obviously makes more sense to report these for May–October. Thus, we will correct this as

BGD
follows: "The mean measured CH4 flux in May–October at the site was  $-4.88 \ \mu$ mol m-2 h-1 (median  $-6.43 \ \mu$ mol m-2 h-1) in 2013 (n=339), and  $-6.46 \ \mu$ mol m-2 h-1 (median  $-5.90 \ \mu$ mol m-2 h-1) in 2014 (n=373), however, the difference was not statistically significant." Furthermore, we noticed that in line 290 we have reported mean air temperature in May–September for different years, and these will be changed to cover May–October, as well. Mean air temperature for May–Oct in 2010–2017 has been 10.0–13.2 C-degrees, in 2013 12.4 and in 2014 12.7 C-degrees.

RC: Fig 4: It would be nice to break these plots up by May-July and August-October observations.

AC: Thank you for this suggestion, we will replace Fig. 4 with a new one with the two seasons separately (see Fig. 2 in this document), and also modify the text accordingly. We also added information about statistically significant differences between the seasons among each vegetation group (plus signs). Moreover, we noticed that there was a mistake in the figure caption of Fig. 4: the triangles are actually medians, whiskers are 25th and 75th percentiles, and asterisks are means.

RC: 361-367: Again, reporting the only the mean is limiting, also report SD (or some other metric of variability) within these sample groups.

AC: We will add SD values.

RC: Fig 5: Was this variability maintained between the early to late summer transition? It would be good to show both groups on this plot, but might make things too cluttered.

AC: We explored the possibility to plot the seasons separately in two different plots (see Fig. 3 in this document). However, it does not provide much new information compared to the modelled flux maps – mainly the wettest sample points shift from emission to uptake of CH4. Also, the figure splitted to two seasons is not very easy to read, as the Referee expected, and moreover the purpose of this figure is to show the spatial variation at sample points. Hence, we think it is best to keep this figure as it is.

BGD
RC: Table 2. It would be very helpful here to report the modeled statistics for both the whole area and at the sample points. Currently it is unclear whether the modeled soil moisture is systematically lower and therefore causing systematic overestimations of CH4 uptake, or if the domain of the entire study area happens to be drier on average leading to a larger estimated CH4 uptake.

AC: We will modify Table 2 as the Referee suggests.

RC: Fig 7: Normalizing the uncertainty at each pixel by its predicted value would help communicate the spatial patterns in the consistency of the RF ensemble output. I would also suggest that the authors add a note on interpreting this uncertainty, which is more of a measurement of the agreement of predictions among multiple RF iterations than the error between predictions and observations like RMSE. I am a major supporter of reporting ensemble uncertainty along with model metrics like RMSE, but the wording can get very confusing!

AC: Thanks for this comment. We will plot the relative uncertainties instead of absolute uncertainties as the Referee suggests, however it is good to note that this will then alter the message conveyed with the figure. We will also modify the text so that it is clear that this figure shows the uncertainty related to the upscaling procedure alone and does not include uncertainties related e.g. to possibly biased training data due to biased sampling locations. For the overall uncertainty, cross-validation metrics such as RMSE are better, like the Referee points out.

RC: 418-421: This is another place where normalizing the uncertainty of the ensemble predictions is useful.

AC: Figure 9 will be modified as the referee suggests, as well as Fig. 7 (see Figs. 4 and 5 in this document), however note that for locations with close to zero fluxes the relative uncertainties will inflate. This is one of the reasons why we opted to plot absolute uncertainties in the first place.

BGD
RC: 422-424: I may have missed it, but I do not remember seeing this approach described in the methods and it is kind of unclear here. I am also confused by what this is supposed to demonstrate.

AC: We acknowledge that the wording on lines 422-425 in the manuscript needs clarification, and this has maybe caused misunderstanding. The idea of the analysis shown in Fig. 10 in the manuscript was to evaluate how many chamber measurement locations were needed to get an accurate estimate of landscape-level flux by only averaging over the measured chamber data without any upscaling with RF. Here the mean upscaled CH4 flux was used as a reference since it accounts for the soil heterogeneity (see above). It was shown that average over 15-20 locations resulted in a similar bias as average over all the chamber measurement locations. This should be useful information for future chamber measurements in similar locations. We will revise the text so that this analysis is clearer and describe the methodology briefly also in Materials and Methods (Sect. 2).

Discussion:

RC: 444: This is unclear. The RF model was just used to estimate spatial distributions of soil moisture, which were then used to predict CH4 flux based on a linear model, correct?

AC: Due to the Referee #1 comment we re-evaluated the modelling approach, and opted to follow reviewer suggestion to use RF model also for CH4 fluxes.

RC: 451-454: I am not sure what these lines are doing in this paragraph. They seem disconnected from the point.

AC: We assume that the discussion on the relationship between vegetation and soil moisture suits here, although, we will try to modify the text to connect them somehow better here and make the text more fluent and effective.

RC: 457: How are these two species different in terms of phenology, growth form, and
root structure? If they are similar, I would hesitate to infer that the vegetation is affecting CH4 flux rather than soil properties other than moisture.

AC: This sentence is not based on just two species, but the vegetation classes. This proposition is very moderate suggestion that there might be other effects than soil moisture, something related to vegetation may affect the CH4 flux. We will see if we can add more information on this.

RC: 469-470: Yes, but variability within point clusters was not communicated to the readers. It would be very useful to include.

AC: It is true that there is some variability in the CH4 fluxes within sample point groups (Fig. 5). We will add a comment about that in the discussion.

RC: 470-472: I do not agree with this. The points created the domain of the training data, so we would expect the model output to be constrained by that domain. Additionally, the mean of the data only tells part of the story. It would be much more useful to compare the distributions of prediction values vs. observations.

AC: We partly agree with this comment. RF model cannot predict outside the range of values in the training data and the same applies to our CH4 upscaling procedure. However, we argue that upscaling can fix biases caused by skewed sampling locations (as long as the training data contains at least some data points at both extremes of the pdf) and hence agreement between the mean values is not trivial. We agree also that mean of the data tells only part of the story, however analysis of mean flux is important if accurate landscape-scale CH4 budgets are strived for. Hence, we would like to keep this part of the story, but comparison of means is important if the target is accurate landscape-scale CH4 budget.

RC: 494-496: This is interesting, what differences in the ecosystems/soil types may account for this?
AC: Unfortunately, we cannot really find a good explanation to this based on the collected data, but think it is relevant to mention.

RC: 503-509: It would be useful to communicate whole plot scale CH4 flux estimates, but net sums and total source and sink strength.

AC: If we understand correctly, the Referee suggests to add discussion about the whole-site mean values presented in Table 2. We agree that the whole-site mean flux and moisture are now getting quite little attention in the discussion, and we will add some discussion on those.

RC: 513-514: This is could also be due to reduced activity of methanogens in deeper soil layers/microsites.

AC: Yes, due to lower soil moisture. At least for the wet areas. We will add this to the same sentence.

RC: 568-571: Yes. But why did this study focus on modeling soil moisture and not directly modeling CH4 flux based on landscape features?

AC: See our answers to the main questions raised.

RC: Conclusions: This section could be filled out more completely. Differences in CH4 flux based on vegetation type was an interesting finding, for example.

AC: Thank you for this comment. We wanted to keep the Conclusion paragraph short. But we can add e.g. the effect of vegetation type here.

References:

Treat, C. C., Bloom, A. A. and Marushchak, M. E.: Nongrowing season methane emissions–a significant component of annual emissions across northern ecosystems, Glob. Chang. Biol., 24(8), 3331–3343, doi:10.1111/gcb.14137, 2018.
Fig. 1. New Fig. 8: Maps of CH4 Flux with modified modelling approach.

---

## Author Response (AR1)

**Point-by-point response to reviewer comments**

**Vainio et al., Biogeosciences**

This point-by-point response to the reviews is based on the earlier responses given during the open discussion.

**Main changes made in the manuscript:**

- The modelling approach for the CH4 fluxes was changed: we model the CH4 fluxes directly with the RF technique and using soil moisture as a driver in the model.
- The Discussion section was completely re-organized and subtitles were added, also some new text was added in the Discussion based on the referee comments.

**Referee #1**

1. The modeling framework

RC: Line 225: How many observations did you have for May-July and August-October? Did you have many measurements from one point in your model (e.g., early May measurement, late July measurement)? Can you be certain that the soil moisture measurements conducted within one study period (e.g. in early May and late July) can be directly compared and used in the same model, even though the soils tend to get drier during the summer? I think it's important that the measurements that you use for your response variable (i.e., soil moisture) are fully comparable with each other.

AC: There were around 6 measurements (median) from each sample point in both of the seasons. Our data is not temporally very comprehensive, and thus we opted to do two static periods. Still, we wanted to take into account that the soil does, in fact, get drier towards autumn. We assume that this two-seasons-strategy is good compromise with the data we have. We actually started the measurements in late May (line 132), so we missed the wettest period in the spring, and the most active measurement period was in June–August (line 135). However, we still ended up having approximately same amount of data for both seasons.

RC: 225: Could you also describe why you decided to use soil moisture as a predictor of fluxes instead of using the different topographic indices directly? Also, did you consider creating a continuous vegetation type raster based on your vegetation classes and the gridded layers for the study domain? This could have been a useful predictor for CH4 fluxes as well.

AC: We re-evaluated our modelling approach and changed the modelling: we model the CH4 fluxes directly with the RF technique and using soil moisture as a driver in the model (see Fig.1 in this document). This change was motivated by the fact that bulk of the criticism from the reviewers was directed towards the CH4 flux modelling approach. The manuscript was modified accordingly.

We considered but did not create a continuous vegetation type raster, because there are many drivers affecting the vegetation, and we did not have data of all such drivers. There is no direct connection between e.g. vegetation and soil moisture (Fig. 4), nor with topography. A continuous vegetation type raster would have probably required more thorough mapping of ground vegetation of the entire area.

RC: 256: Why didn't you use the similar framework that you used for soil moisture to predict CH4 fluxes with soil moisture? You could have created a RF (or a GLM/GAM or some other) model with the measured

soil moisture as a predictor, and then used that model to predict fluxes across the landscape using the predicted soil moisture. And this could have been repeated over the different bootstrapped soil moisture maps to get CH4 flux uncertainty map as well.

AC: See the previous response, we changed our modelling approach to follow this referee suggestion.

RC: 240:  Could you add the response graphs (partial dependence plots) describing the relationship of these indices and soil moisture to the Appendix?

AC: These were added to the Appendix.

**2. Description of the model performance**

RC: Line 20: Somewhere here I would add a sentence about how the statistical models performed, and how reliable your results are.

AC: We added something more about the model to the abstract.

RC: 377: I would be interested to see a scatterplot of the observed and predicted (CV) fluxes to see how well the model predicts high and low soil moisture values. Same applies to CH4 flux (line 406).

AC: We added the scatterplot showing cross-validation results to the Appendix.

RC: 542: Somewhere in the Discussion you should also discuss how well your upscaling performed compared to previous studies. What are the main uncertainties, and how can these uncertainties be reduced? What predictors are you missing? How about other RS-derived indices, such as NDVI?

AC: According to our current impression, there are not many published studies trying to upscale CH4 fluxes from forest stand like ours. The previous study of (Sundqvist et al. 2015) used simple soil wetness and temperature relationship to upscale CH4 fluxes, whereas Kaiser et al., (2018) and Warner et al., (2019) studied more southern ecosystems. Furthermore, comparisons of e.g. cross-validation results are hampered by different cross-validation techniques used in different studies. In this manuscript, we utilized distance-blocked cross-validation since it is argued to produce more realistic estimates of cross-validation metrics than other techniques (Roberts et al., 2017). For instance, with traditional leave-one-out cross-validation the predictive performance of our RF model would seemingly improve (e.g. $r^2$ increase from 0.51 and 0.26 to 0.67 and 0.56 for May–July and August–October, respectively). Therefore, we argue that direct comparison of cross-validation metrics between studies using different cross-validation strategies is not feasible. We added a section to the discussion on how these predictors are missing in upscaling exercises. The scale where our measurements are carried out cannot be directly applied to similar spatial scale than remote sensing methods that is used to estimate NDVI, despite the rapid development of satellite products.

RC: Title: I would consider adding the word "statistical modelling" somewhere in the title

AC: We changed the title to "Topography-based statistical modelling reveals …"

RC: Line 19: I would say "using digital elevation model-derived topographic indices" instead of "topography"

AC: We modified this as suggested.

RC: 28: I was a bit surprised to see this methodological suggestion as a final sentence concluding your study. I would consider changing this to something more broader, e.g. to the sentence on line 565-572.

AC: We modified this as suggested.

RC: 71: "Large amount of measurement points" is a rather subjective statement as for some people this might mean hundreds or thousands of observations. Maybe define the rough amount of measurement points instead, and mention that this is more than has previously been used.

AC: We clarified this sentence in the following way: "In this study, we used relatively high number of measurement points (60 points on an area of ca. 10 ha) in order to fully cover the small-scale spatial variability in the CH4 flux and its driving forces. Similar type of studies using chamber measurements are rarely based on more than 20 measurement points."

RC: 75: With one driving parameter (i.e. soil moisture), right? You didn't have many driving parameters to make the upscaled CH4 flux map?

AC: Yes, originally we used soil moisture only. However, due to the Referee #1 comment we re-evaluated this approach and based on this re-evaluation opted to follow reviewer suggestion to use RF model also for CH4 fluxes.

RC: 76: But what about Kaiser et al., 2018?

AC: We modified the text to: "Only a few studies (Kaiser et al., 2018; Warner et al., 2019) have applied similar approach, of which Kaiser et al. (2018) at a boreal coniferous forest, emphasizing the novelty of this study."

RC: 105: You could add an index map to this figure showing e.g. the location of Hyytiälä, too.

AC: In the end, we opted not to add a new map.

RC: 210: I would describe these gridded layers in their own paragraph, similar to the other environmental measurements, and dedicated this one to the models only.

AC: Good suggestion, we separated these under a new subheading.

RC: 220: Could you provided a little bit more information about what parameters were chosen for the different indices? For example, TWI can change quite a bit depending on what parameters you use in the calculation.

AC: We added more information on these to the methods.

RC: 370: This figure could be moved to the supplementary – it's not so useful for the reader because there are so many different points.

AC: We think that this figure gives a nice overview of the measured CH4 flux and its variation at the measurement points, and it is rather easy to see the spatial variation from this figure. (E.g. Fig. 4 is not giving this information, and a table would be more difficult to read.)

RC: 400: This is just an idea, but you could also replace these two maps by a map that describes the mean summer soil moisture and a map that describes its change over the growing season. It might be easier for the reader to spot the areas that are drying this way.

AC: This is an interesting thought and good to consider. However, we think that it might be more difficult for readers to understand what was done in this paper based on such a figure.

RC: 431: I would use the same color scheme that you used in Fig. 8 for the Fig. 9 as well, to make sure that you are using different color schemes for soil moisture and CH4 fluxes.

AC: Diverging colormaps (as the one used in Fig. 8) are suitable for data sets containing negative and positive values, because this way it is possible to emphasize the difference from zero. However, based on

Referee #2 comment, we modified Figs. 7 & 9 to show relative uncertainties, and as a result Fig. 9 colormap was changed.

RC: 440: The discussion is rather long and without subtitles it is a little bit hard to follow. Could you consider adding a few subtitles and structuring it according to your main aims of the study (spatial variation, drivers and upscaling, hot spots)?

AC: We divided the discussion under subtitles by re-organizing the text.

RC: 560: Again, I was a bit surprised to see this discussion here as it was not motivated in your introduction or it wasn't one of your main aims of the paper. Maybe include it to the introduction or remove it completely?

AC: This is a good remark. We added shortly to the introduction that previously usually fewer measurement points have been used in soil chamber CH4 measurements, with the assumption that they are representative for a larger area.

RC: 567: If you want to discuss the sampling strategy, I would provide some more details here. E.g., how should the sample points be selected (e.g. systematic grid, gradient, random sampling, researcher-defined)? What is the number of temporal replicates required to understand spatiotemporal variability in this system? Further, in the abstract you mention that capturing the environmental variability requires 15-20 sample points. But do you think using statistical methods (e.g. random forest) with 15-20 points is reliable?

AC: Thank you for this comment. We added shortly in this paragraph about the selection of sample points: in our opinion, e.g. the elevation maps would be useful when selecting the sample points. With this study we were not able to reveal more high-frequency temporal variability, but we can add a sentence here that more frequent measurements would be needed for that – ideally it would require automatic chambers measuring e.g. once per day, or at minimum manual measurements every week.

The idea of the analysis shown in Fig. 10 in the manuscript was to evaluate how many chamber measurement locations were needed to get an accurate estimate of landscape-level flux by only averaging over the measured chamber data without any upscaling with RF. This was clarified in the manuscript, as this was also commented by Referee #2.

References:

Kaiser, K. E., McGlynn, B. L. and Dore, J. E.: Landscape analysis of soil methane flux across complex terrain, Biogeosciences, 15(10), 3143–3167, doi:10.5194/bg-15-3143-2018, 2018.

Roberts, D. R., Bahn, V., Ciuti, S., Boyce, M. S., Elith, J., Guillera-Arroita, G., Hauenstein, S., Lahoz-Monfort, J. J., Schröder, B., Thuiller, W., Warton, D. I., Wintle, B. A., Hartig, F. and Dormann, C. F.: Cross-validation strategies for data with temporal, spatial, hierarchical, or phylogenetic structure, Ecography (Cop.)., 40(8), 913–929, doi:10.1111/ecog.02881, 2017.

Sundqvist, E., Persson, A., Kljun, N., Vestin, P., Chasmer, L., Hopkinson, C. and Lindroth, A.: Upscaling of methane exchange in a boreal forest using soil chamber measurements and high-resolution LiDAR elevation data, Agric. For. Meteorol., 214–215, 393–401, doi:10.1016/j.agrformet.2015.09.003, 2015.

Warner, D. L., Guevara, M., Inamdar, S. and Vargas, R.: Upscaling soil-atmosphere CO2 and CH4 fluxes across a topographically complex forested landscape, Agric. For. Meteorol., 264, 80–91, doi:10.1016/j.agrformet.2018.09.020, 2019.

**Referee #2**

RC: There are two larger general comments: 1) Concerning the upscaling method:

AC: We re-evaluated our modelling approach based on these Referee comments, and changed the modelling to follow the suggestion by Referee #1: we model the CH4 fluxes directly with the RF technique and using soil moisture as a driver in the model.

RC: On top of this, there is never a discussion of why the approach of modeling soil moisture and then CH4 flux is advantageous. Furthermore, the authors dedicate a large portion of this manuscript to the upscaling exercise, but barely, if at all, discuss whole plot scale fluxes. It would be interesting to hear how much the estimated net CH4 sources offset the plot level sink between the two time periods, and how uncertain their plot level fluxes are. After all, the primary purpose of upscaling is not to accurately predict CH4 flux at every individual point, it is to enhance our predictive capability of large-scale CH4 exchange in a way that reflects soil heterogeneity.

AC: See the previous response, we changed the modelling approach.

We agree with the referee that the primary purpose of upscaling is to get an accurate estimate of landscape fluxes and not at individual points. We revised the text so that this message gets across to the reader clearly, and added some discussion on the whole area flux and the modelling. This was actually one of the main reasons why the analysis on chamber location bias was done (Fig. 10). Typically, mean of CH4 fluxes observed at a handful of chamber locations is reported and considered as representative of ecosystem CH4 exchange, however, this neglects any bias stemming from non-representative sampling locations. By using mean upscaled CH4 flux as a reference, we were able to show that 15–20 randomly selected chamber measurement locations (out of 60 locations available) were able to produce as accurate estimate of the landscape CH4 flux as averaging over all the chamber data. This information should prove useful when designing future chamber measurement campaigns in similar locations aiming to achieve accurate landscape-level flux estimates without upscaling with e.g. machine learning techniques.

RC: It would be more useful to randomly subsample the flux observations and build soil moisture-CH4 relationships from the random subsets. Then see how upscaled fluxes based on these relationships compare to the predictions made using the whole dataset.

AC: We acknowledge that the wording on lines 422-425 in the original manuscript needed clarification, and this has maybe caused misunderstanding. The idea of the analysis shown in Fig. 10 in the manuscript was to evaluate how many chamber measurement locations were needed to get an accurate estimate of landscape-level flux by only averaging over the measured chamber data without any upscaling with RF. Here the mean upscaled CH4 flux was used as a reference since it accounts for the soil heterogeneity (see above). It was shown that average over 15-20 locations resulted in a similar bias as average over all the chamber measurement locations. This should be useful information for future chamber measurements in similar locations. We revised the text so that this analysis is clearer and describe the methodology briefly also in Materials and Methods.

RC: The authors should also report their modeled CH4 predicted fluxes for pixels corresponding to the sample sites, which would help explain whether differences in upscaled fluxes are caused by a model bias or because of the heterogeneity of the predictor variable domain.

AC: Thank you for this comment, we added the modelled flux of the sample point locations (pixels) to Table 2.

**Specific Comments**

RC: Abstract: If possible, add some descriptive statistics (i.e. mean, min, max) for what CH4 fluxes were observed in each season.

AC: These were added.

RC: 21: The wording "as well as on the related ground vegetation" is confusing to me.

AC: Corrected "as well as from the related ground vegetation".

RC: 33-35: I do not believe that this is the current paradigm. Observed methane fluxes are the net sum of both opposing processes occurring in the soil.

AC: While it is the prevailing paradigm that the availability of oxygen mainly controls these processes in nature, and thus in general upland soils are no favourable place for CH4 production, there are possibilities for CH4 production taking place in these soils too, as we state in the next paragraph. The recent study of Treat et al. (2018) reported that during non-growing season upland soils can be a significant source of methane although they are generally considered as sinks during the growing season.

RC: 40: This is not an instant effect, however, which has implications on the influence of both total soil moisture and its temporal variability. It would be useful to note this here.

AC: We agree that the soils do not turn immediately sources of CH4 after inundation, but there are time lags between these processes. We added the following sentence: "However, there are likely notable time lags between the start of inundation and methanogenesis, complicating the analyses of dependencies between these processes.".

RC: 53-56: I think this is a great point, but I am confused why it is in this paragraph.'

AC: Thank you for this constructive comment but our impression is that this is logically introduced as it is.

RC: 57-62: This paragraph is important but it could be written more clearly. Are the authors trying to say that we often consider ecosystem fluxes in large-scale models but have not adequately accounted for heterogeneous sources/sinks within the ecosystem (which likely respond to environmental changes differently)?

AC: Yes, the sources and sinks within the ecosystems are not adequately known or accounted. We added a sentence in this paragraph, clarifying this.

**Methods:**

RC: 152: Is there an explanation for why one measurement would be so large?

AC: The sample point was located on small water pond, where the water table level was most of the time above the peat surface. The CH4 emissions from this sample point were at the same level with typical peatland emissions. One possible explanation for the highest emission would be ebullition.

RC: 177: It would be good to provide a +/- range for what types of temperature variation were observed here as justification.

AC: We added the range of average temperature at the sample points in late June 2013, when the temperature measurements at all the sample points were started.

RC: 220: Was the TWI then resampled/interpolated or left at coarse resolution? Additionally, I would hesitate to say that the TWI is "not accurate" at fine scales since it is simply a statistical metric and not a

measurement of anything. It does, however, have limited application for estimating soil moisture on very high resolution DEMs because the metric itself is very sensitive to surface microtopography and noise.

AC: Yes, TWI was then interpolated with bilinear interpolation to the finer grid (5x5). We clarified the text.

RC: 225-242: I believe this paragraph could be rewritten to be better related to this study. A lot of the information on the inner workings of the RF algorithm can be condensed with an appropriate citation and a brief note on the advantages/disadvantages of RF over simpler techniques like multiple regressions. More information on why the model parameters (ntree = 300, mtry = 2) and predictor variables were selected would be helpful.

AC: We believe that some description is needed since the method is not necessarily very familiar to the chamber flux measurement community. Minimum number of observations per tree leaf was set to 2, due to limited amount of data. Small number for this parameter helps when trying to capture dependencies at both ends of the training data distribution, especially when using limited dataset. Amount of trees in the RF model (ntree) was initially set to 300 based on prior experience with RF and value for this hyperparameter did not significantly influence results. However, the number of variables randomly sampled as candidates at each split (mtry) was not changed from its default value (one third of the total number of variables). The predictor variables were selected based on Spearman's rank correlation coefficient.

**Results:**

RC: 304-306: Having S.D. values or some indicator of variability in soil moisture besides these means would be helpful here and in other parts of the results section.

AC: We added the SD values in text with the reported mean values.

RC: 326-327:  What was the data of the highest emission outlier? It could be nice to see where it and other CH4 measurements fall on the time series graphs above.

AC: The CH4 highest emission (1080 µmol m−2 h−1) was measured on 5 June 2013. We added the date of the measurement in the text (line 326). It is unfortunately difficult to present an approachable timeseries plot of the measured CH4 fluxes due to high variation (large emissions vs. other data) – furthermore, we think there is no need to add any more figures to this paper.

RC: 333-334 and Fig 3: This is unclear to me. Is this a temporally static correlation between the mean of all CH4 fluxes at each point and the mean of soil moisture at each point?

AC: Yes.

RC: 339: Is "September" supposed to be "October" here?

AC: Thank you for pointing this out. These were actually calculated for May–Sep, and was corrected for May–October as follows: "The mean measured CH4 flux in May–October at the site was −4.88 µmol m−2 h−1 (median −6.43 µmol m−2 h−1) in 2013 (n=339), and −6.46 µmol m−2 h−1 (median −5.90 µmol m−2 h−1) in 2014 (n=373), however, the difference was not statistically significant."

Furthermore, we noticed that in line 290 we had reported mean air temperature in May–September for different years, and these were also changed to cover May–October, as well. Mean air temperature for May–Oct in 2010–2017 has been 10.0–13.2 C-degrees, in 2013 12.4 and in 2014 12.7 C-degrees.

RC: Fig 4:  It would be nice to break these plots up by May-July and August-October observations.

AC: Thank you for this suggestion, we replaced Fig. 4 with a new one with the two seasons separately, and modified the text accordingly. We also added information about statistically significant differences between the seasons among each vegetation group (plus signs). Moreover, we noticed that there was a mistake in

the figure caption of Fig. 4: the triangles are actually medians, whiskers are 25$^{th}$ and 75$^{th}$ percentiles, and asterisks are means – these were corrected in the caption.

RC: 361-367: Again, reporting the only the mean is limiting, also report SD (or some other metric of variability) within these sample groups.

AC: We added SD values.

RC: Fig 5: Was this variability maintained between the early to late summer transition? It would be good to show both groups on this plot, but might make things too cluttered.

AC: We explored the possibility to plot the seasons separately in two different plots (see below). However, it does not provide much new information compared to the modelled flux maps – mainly the wettest sample points shift from emission to uptake of CH4. Also, the figure splitted into two seasons is not very easy to read, as the Referee expected, and moreover the purpose of this figure is to show the spatial variation at sample points. Hence, we think it is best to keep this figure as it is.

[Figure]

Fig. 5 of the manuscript splitted into two seasons.

RC: Table 2. It would be very helpful here to report the modeled statistics for both the whole area and at the sample points. Currently it is unclear whether the modeled soil moisture is systematically lower and therefore causing systematic overestimations of CH4 uptake, or if the domain of the entire study area happens to be drier on average leading to a larger estimated CH4 uptake.

AC: We added the modelled values at the sample point locations to Table 2.

RC: Fig 7: Normalizing the uncertainty at each pixel by its predicted value would help communicate the spatial patterns in the consistency of the RF ensemble output. I would also suggest that the authors add a note on interpreting this uncertainty, which is more of a measurement of the agreement of predictions among multiple RF iterations than the error between predictions and observations like RMSE. I am a major

supporter of reporting ensemble uncertainty along with model metrics like RMSE, but the wording can get very confusing!

AC: Thanks for this comment. We plotted the relative uncertainties instead of absolute uncertainties as the Referee suggests, however it is good to note that this will then alter the message conveyed with the figure. We also modified the text so that it is clear that this figure shows the uncertainty related to the upscaling procedure alone, and does not include uncertainties related e.g. to possibly biased training data due to biased sampling locations. For the overall uncertainty, cross-validation metrics such as RMSE are better, like the Referee points out.

RC: 418-421: This is another place where normalizing the uncertainty of the ensemble predictions is useful.

AC: Figure 9 was modified as the referee suggests, as well as Fig. 7, however note that for locations with close to zero fluxes the relative uncertainties will inflate. This is one of the reasons why we opted to plot absolute uncertainties in the first place.

RC: 422-424: I may have missed it, but I do not remember seeing this approach described in the methods and it is kind of unclear here. I am also confused by what this is supposed to demonstrate.

AC: We acknowledge that the wording on lines 422–425 in the original manuscript needed clarification, and this has maybe caused misunderstanding. The idea of the analysis shown in Fig. 10 in the manuscript was to evaluate how many chamber measurement locations were needed to get an accurate estimate of landscape-level flux by only averaging over the measured chamber data without any upscaling with RF. Here the mean upscaled CH4 flux was used as a reference since it accounts for the soil heterogeneity (see above). It was shown that average over 15–20 locations resulted in a similar bias as average over all the chamber measurement locations. This should be useful information for future chamber measurements in similar locations. We revised the text so that this analysis is clearer and describe the methodology briefly also in Materials and Methods.

**Discussion:**

RC: 444: This is unclear. The RF model was just used to estimate spatial distributions of soil moisture, which were then used to predict CH4 flux based on a linear model, correct?

AC: Due to the Referee #1 comment we re-evaluated the modelling approach, and opted to follow reviewer suggestion to use RF model also for CH4 fluxes.

RC: 451-454: I am not sure what these lines are doing in this paragraph. They seem disconnected from the point.

AC: We assume that the discussion on the relationship between vegetation and soil moisture suits here, although, we have now completely rearranged the Discussion section.

RC: 457: How are these two species different in terms of phenology, growth form, and root structure? If they are similar, I would hesitate to infer that the vegetation is affecting CH4 flux rather than soil properties other than moisture.

AC: This sentence is not based on just two species, but the vegetation classes. This proposition is very moderate suggestion that there might be other effects than soil moisture, something related to vegetation may affect the CH4 flux.

RC: 469-470: Yes, but variability within point clusters was not communicated to the readers. It would be very useful to include.

AC: It is true that there is variation in the CH4 fluxes within sample point groups (Fig. 5). We add a sentence about that in the Discussion (lines 492–494).

RC: 470-472: I do not agree with this. The points created the domain of the training data, so we would expect the model output to be constrained by that domain. Additionally, the mean of the data only tells part of the story. It would be much more useful to compare the distributions of prediction values vs. observations.

AC: We partly agree with this comment. RF model cannot predict outside the range of values in the training data and the same applies to our CH4 upscaling procedure. However, we argue that upscaling can fix biases caused by skewed sampling locations (as long as the training data contains at least some data points at both extremes of the pdf) and hence agreement between the mean values is not trivial. We agree also that mean of the data tells only part of the story, however analysis of mean flux is important if accurate landscape-scale CH4 budgets are strived for. Hence, we would like to keep this part of the manuscript as it is, but included a sentence about the fact that mean tells only part of the story, but comparison of means is important if the target is accurate landscape-scale CH4 budget (line 572).

RC: 494-496: This is interesting, what differences in the ecosystems/soil types may account for this?

AC: Unfortunately, we cannot really find a good explanation to this based on the collected data, but think it is relevant to mention.

RC: 503-509:  It would be useful to communicate whole plot scale CH4 flux estimates, but net sums and total source and sink strength.

AC: If we understand correctly, the Referee suggests to add discussion about the whole-site mean values presented in Table 2. We tried to emphasize the main message in the discussion.

RC: 513-514:  This is could also be due to reduced activity of methanogens in deeper soil layers/microsites.

AC: Yes, due to lower soil moisture. At least for the wet areas. We added this to the same sentence.

RC: 568-571: Yes. But why did this study focus on modeling soil moisture and not directly modeling CH4 flux based on landscape features?

AC: See our answers to the main questions raised.

RC: Conclusions: This section could be filled out more completely. Differences in CH4 flux based on vegetation type was an interesting finding, for example.

AC: Thank you for this comment. We wanted to keep the Conclusion paragraph short.

References:

Treat, C. C., Bloom, A. A. and Marushchak, M. E.: Nongrowing season methane emissions–a significant component of annual emissions across northern ecosystems, Glob. Chang. Biol., 24(8), 3331–3343, doi:10.1111/gcb.14137, 2018.

---

## Referee Report (RR1)

Vainio et al. have made major changes to their manuscript which have greatly improved it, but there are still a few minor points that should be clarified.

I have only one major point:

I had a question on your previous manuscript version on line 225 (the question about the number and distribution of measurements) which I am still not fully understanding. I am not worried about the temporal or spatial comprehensiveness of your measurements, but it would be important to understand when the soil moisture and CH4 flux measurements that were used to train each model (May-July or Aug-Oct models) were taken and how the measurement date could influence your model performance. For example, ideally, you would do your soil moisture measurements across all the sampling locations on the same day, to make sure your measurements are representing the same conditions, and use this information to train your soil moisture model. Now I think your measurements are distributed quite randomly throughout the model period (either May-July or Aug-Oct), making them less comparable with each other. For example, you might have measured one location on a sunny day in early June and another one during a cold and rainy day in late June. Or, you might have measured one sampling location primarily in May, whereas you measured another location only in July. Even though the topographic position, soil texture, and vegetation conditions would be identical, and consequently soil moisture of these two locations should be the same if they would be measured at the same time, the location sampled only in July might suggest that the soil is drier than the other location because it tends to dry out during the summer. What I mean is that some part of the unexplained variability in your soil moisture/CH4 flux models might be related to the measurement time (date) and how warm and rainy it has been before it. And your current predictors do not consider this variability at all since they describe static topographic properties. I would either try to test how the measurement date or air temperature/precipitation conditions prior/during your measurement explain soil moisture or CH4 flux to check how much this could explain the performance issues in your models, or discuss this as a potential reason for the fact that a relatively large amount of variability in soil moisture/CH4 flux remained unexplained somewhere in discussion. You are kind of discussing this on line 620->, but I think you should also mention the comparability issues of your measurements somewhere, if I have understood the sampling scheme correctly.

Minor points:

29: "which was enough" sounds a bit strange to me

30: CH4 -> $CH_4$

31: "upscaling predicted stronger CH4 uptake" -> do you have an idea why? Because you might have sampled more wetter environments than what their aerial extent truly was? Maybe this could be discussed briefly in the discussion.

33-35: I still think these points come a bit out of the blue here in the abstract because they haven't been mentioned before. Maybe you could try to link this sampling strategy comment more specifically (but shortly) to your results listed in the abstract, e.g. to the ones related to the differences in measured vs. predicted fluxes (see my comment above)? The main text mentions the sampling strategy in the introduction and discussion, which is great, but I'm just trying to assure that the reader understands the abstract as well as possible too.

35: "…, and the measured fluxes…" seems a bit redundant to be. Doesn't it always make sense to link fluxes to their environmental drivers?

213: distribution was non-normal

237: "The calculations were made with TopoToolbox (Schwanghart and Kuhn, 2010)." -> can be removed, since this is already mentioned on line 234

242: What was the spatial resolution of the other gridded data sets mentioned on lines 228?

252: You could report here how many measurements and sampling locations you used in these two models. I assume that you had data from 60 sampling locations in all models, but it's unclear to me how many observations you had in total (i.e. what was the sample size in your models).

269: I'm not sure I understand the beginning of this sentence/ whether it's formulated correctly: "In this method, one RF model is developed for each sample point…". To me it says that you developed a model for each individual sample point (i.e. model sample size n=1). Maybe mention that the validation data came from one sample point/you predicted the model developed with the training data to each individual sample point or something similar.

278: Perhaps mention here how you calculated uncertainty. Many studies also use a prediction interval of e.g. $2.5^{th}$-$97.5^{th}$ percentiles. Why did you choose to use standard deviations?

279: I think your uncertainty estimate also describes the effect of the distribution of the sampling points.

281: Aalto et al. (2018)

283: Do you think it's problematic that your soil moisture predictions are not entirely independent from the topographic indices that you use as predictors of fluxes as well? Ideally, you'd use entirely independent data sets to train your CH4 flux models. Now your soil moisture predictions already have some information about the topographic indices too.

438: I'd mention somewhere here that your soil moisture and CH4 flux models seemed to have issues particularly with small and large fluxes, which were over- and underestimated, respectively. This might influence your average predicted fluxes too. E.g., if the area covered by dry conditions which are suitable for net CH4 uptake is high compared to the wet CH4 emitting patches, maybe your predictions are underestimating the net uptake and the difference between measured and predicted fluxes could actually be even greater?

537: I would not repeatedly mention the acronym in the discussion SW-W-3

538: Do you have the data about water table depth somewhere?

547: Or is it possible that SW-W-3 was measured after a rainy day and other on drier conditions?

566-599: Could this discussion be its own section? Now that it's listed under upscaling, I start to think again about the limitations in using 15-20 sampling points to train a random forest model, even though I understand that this discussion is not related to that.

582: You could add a few references here. Also, I guess quite frequently mean flux is calculated for each vegetation type, and then a spatially representative mean flux is calculated based on the spatial extent of the vegetation types? Or is that still rare in boreal forest CH4 flux studies? This method of course has issues related to the vegetation type map used, so I think your method is better!

584: "spatially modelled CH4 flux" – are you referring to the modelled estimate for the whole area?

612: But how about vegetation indices derived from WorldView, Planet, or Sentinel-2 that have a pixel resolution of 2-10 meters?

613-619: A large part of this paragraph could actually be located in Section 4.1. which focuses on the drivers.

---

## Author Response (AR2)

Referee:

I had a question on your previous manuscript version on line 225 (the question about the number and distribution of measurements) which I am still not fully understanding. I am not worried about the temporal or spatial comprehensiveness of your measurements, but it would be important to understand when the soil moisture and CH4 flux measurements that were used to train each model (May-July or Aug-Oct models) were taken and how the measurement date could influence your model performance. For example, ideally, you would do your soil moisture measurements across all the sampling locations on the same day, to make sure your measurements are representing the same conditions, and use this information to train your soil moisture model. Now I think your measurements are distributed quite randomly throughout the model period (either May-July or Aug-Oct), making them less comparable with each other. For example, you might have measured one location on a sunny day in early June and another one during a cold and rainy day in late June. Or, you might have measured one sampling location primarily in May, whereas you measured another location only in July. Even though the topographic position, soil texture, and vegetation conditions would be identical, and consequently soil moisture of these two locations should be the same if they would be measured at the same time, the location sampled only in July might suggest that the soil is drier than the other location because it tends to dry out during the summer. What I mean is that some part of the unexplained variability in your soil moisture/CH4 flux models might be related to the measurement time (date) and how warm and rainy it has been before it. And your current predictors do not consider this variability at all since they describe static topographic properties. I would either try to test how the measurement date or air temperature/precipitation conditions prior/during your measurement explain soil moisture or CH4 flux to check how much this could explain the performance issues in your models, or discuss this as a potential reason for the fact that a relatively large amount of variability in soil moisture/CH4 flux remained unexplained somewhere in discussion. You are kind of discussing this on line 620->, but I think you should also mention the comparability issues of your measurements somewhere, if I have understood the sampling scheme correctly.

Authors:

Yes, the referee is right. All the chamber locations were not measured during the same day, but (nearly) all the sample points were always measured during a 5-day-period, and these measurement rounds were repeated approximately every third week in May–October. On average, each sample point was measured every 22 days in May–October. This information was added on lines 148–151. Still, there were differences between the timing of measurements at different locations, which is somewhat inevitable when sampling many locations manually with limited workforce. We tried to overcome the effect of this issue on upscaling by doing two static predictions with the RF model, where the RF models were trained with temporally averaged data. However, despite the averaging, there may have been some leftover variability between the temporal means e.g. due to sampling some locations more during rainy days and others more during hot days. The apparent spatial variability caused by unsynchronised sampling is something that cannot be explained with the topographic properties as the referee points out. This apparent variability would decrease the performance of RF model in explaining the observed spatial variability in soil moisture/CH4 flux. We will add a note on this in the discussion section of the manuscript.

Minor points:

R: 29: "which was enough" sounds a bit strange to me → A: Removed.

R: 30: CH4 -> CH4 → A: corrected

R: 31: "upscaling predicted stronger CH4 uptake"-> do you have an idea why? Because you might have sampled more wetter environments than what their aerial extent truly was? Maybe this could be discussed briefly in the discussion. → A: Yes, our interpretation of these results is that we sampled more wetter environments than what their true areal extent was, and upscaling rectified this sampling bias.

R: 33-35: I still think these points come a bit out of the blue here in the abstract because they haven't been mentioned before. Maybe you could try to link this sampling strategy comment more specifically (but shortly) to your results listed in the abstract, e.g. to the ones related to the differences in measured vs. predicted fluxes (see my comment above)? The main text mentions the sampling strategy in the introduction and discussion, which is great, but I'm just trying to assure that the reader understands the abstract as well as possible too. → A: We added a sentence about this analysis in the abstract, to the methods part.

R: 35: "…, and the measured fluxes…" seems a bit redundant to be. Doesn't it always make sense to link fluxes to their environmental drivers? → A: We rewrote the sentence, so that it doesn't sound like it's not usually done.

R: 213: distribution was non-normal → A: corrected

R: 237: "The calculations were made with TopoToolbox (Schwanghart and Kuhn, 2010)." -> can be removed, since this is already mentioned on line 234 → A: We removed the reference, but probably good to keep the information that also these calculations were made with TopoToolbox, and not only the DEM pre-processing, which is previously mentioned.

R: 242: What was the spatial resolution of the other gridded data sets mentioned on lines 228? → A: They were 16 x 16 m resolution. This is now added to the text, lines 232–234.

R: 252: You could report here how many measurements and sampling locations you used in these two models. I assume that you had data from 60 sampling locations in all models, but it's unclear to me how many observations you had in total (i.e. what was the sample size in your models). → A: Yes, we used all the 60 sample points both in May–July and Aug–Oct. We used the sample point averages for the modelling ((on average 7 and 5 measurements for each sample point location during May–July and August–October, respectively)), while the total number of measurements for both seasons were 392 and 320. We clarified this on lines 259–261, and also added the total numbers of measurements for both seasons to line 375.

R: 269: I'm not sure I understand the beginning of this sentence/whether it's formulated correctly: "In this method, one RF model is developed for each sample point…". To me it says that you developed a model for each individual sample point (i.e. model sample size n=1). Maybe mention that the validation data came from one sample point/you predicted the model developed with the training data to each individual sample point or something similar. → A: We agree, the wording was a bit misleading. What we mean is that one RF model was developed for each chamber measurement location, not for each observed data point. We will try to clarify this in the text.

R: 278: Perhaps mention here how you calculated uncertainty. Many studies also use a prediction interval of e.g. 2.5th-97.5th percentiles. Why did you choose to use standard deviations? → A: We added the notion that the uncertainty was estimated as standard deviation over the ensemble. If we assume gaussian distribution for the values predicted in the ensemble then the prediction interval (e.g. 2.5th–97.5th percentiles) can be directly calculated from the reported standard deviations. The use of standard deviations was just a practical choice.

R: 279: I think your uncertainty estimate also describes the effect of the distribution of the sampling points. → A: The referee might be right. We added this to the text.

R: 281: Aalto et al. (2018) → A: corrected

R: 283: Do you think it's problematic that your soil moisture predictions are not entirely independent from the topographic indices that you use as predictors of fluxes as well? Ideally, you'd use entirely independent data sets to train your CH4 flux models. Now your soil moisture predictions already have some information about the topographic indices too. → A: Yes, we agree, the variables used in CH4 models are to some degree related to each other. However, note that in the RF model training phase the measured soil moistures were used as predictors for CH4 fluxes, and only in the upscaling phase (i.e. when the RF model is used to predict CH4 fluxes in the study domain) predicted soil moistures (directly related to the topographic indices) were used. In general, the RF algorithm is robust towards redundant predictors when the aim is only to predict the response variable (soil moisture and CH4 flux in our case) as accurately as possible. Furthermore, this approach was suggested by a referee during the previous review round and we opted to follow hers/his suggestion.

R: 438: I'd mention somewhere here that your soil moisture and CH4 flux models seemed to have issues particularly with small and large fluxes, which were over-and underestimated, respectively. This might influence your average predicted fluxes too. E.g., if the area covered by dry conditions which are suitable for net CH4 uptake is high compared to the wet CH4 emitting patches, maybe your predictions are underestimating the net uptake and the difference between measured and predicted fluxes could actually be even greater? → A: Good suggestion, we added a note on this on lines 446–447.

R: 537: I would not repeatedly mention the acronym in the discussion SW-W-3 → A: Yes, SW-W-3 is mentioned several times in this section of the text, however the IDs for the chamber locations are given so that it can be shortly and clearly indicate which point is meant, and hence repetition is to some degree inevitable.

R: 538: Do you have the data about water table depth somewhere? → A: Unfortunately not.

R: 547: Or is it possible that SW-W-3 was measured after a rainy day and other on drier conditions? → A: No, this was not the case. That sample point had a water table level above the surface for most of the time.

R: 566-599: Could this discussion be its own section? Now that it's listed under upscaling, I start to think again about the limitations in using 15-20 sampling points to train a random forest model, even though I understand that this discussion is not related to that. → A: Yes, we followed this suggestion and added a new section '4.3 Representativeness of sample point locations'.

R: 582: You could add a few references here. Also, I guess quite frequently mean flux is calculated for each vegetation type, and then a spatially representative mean flux is calculated based on the spatial extent of the vegetation types? Or is that still rare in boreal forest CH4 flux studies? This method of course has issues related to the vegetation type map used, so I think your method is better! → A: Yes, we think that these type of upscaling methods are quite new, and this mean flux has been the traditional way.

R: 584: "spatially modelled CH4 flux"– are you referring to the modelled estimate for the whole area? → A: Yes, we clarified this accordingly.

R: 612: But how about vegetation indices derived from WorldView, Planet, or Sentinel-2 that have a pixel resolution of 2–10 meters? → A: Yes, these remote-sensing observations have good resolution but to our understanding it is not possible to extract the forest floor signal from these observations. We tried to clarify the sentence a bit more.

R: 613-619: A large part of this paragraph could actually be located in Section 4.1. which focuses on the drivers. → Yes, maybe so, this is partly about the drivers, but here they are discussed as they are linked to the modelling as well.